# Transcriptomic changes associated with infection of *Nicotiana benthamiana* plants with tomato ringspot virus (genus *Nepovirus*) during the acute symptomatic stage and after symptom recovery

**Dinesh Babu Paudel**[ID][1¤*]**, Ana Priscilla Montenegro Alonso**[1]**, Joan Chisholm**[2]**, Huogen Xiao**[2]**, Hélène Sanfaçon**[ID][2*]

**1** Saskatoon Research and Development Centre, Agriculture and Agri-Food Canada, Saskatoon, Saskatchewan, Canada, **2** Summerland Research and Development Centre, Agriculture and Agri-Food Canada, Summerland, British Columbia, Canada

¤ Current address: Department of Plant Sciences, University of Saskatchewan, Saskatoon, Saskatchewan, Canada

* bdnesh@gmail.com (DBP); helene.sanfacon@agr.gc.ca (HS)

## Abstract

We have characterized the transcriptome of *Nicotiana benthamiana* plants infected with tomato ringspot virus (ToRSV), a nepovirus. We analyzed two different stages of infection: the acute systemic symptomatic stage and the symptom recovery stage in which young leaves emerge without visible symptoms. In agreement with previous observations, we note a similar concentration of viral RNAs in symptomatic and recovered leaves. Extensive reprogramming of the plant transcriptome was observed in symptomatic leaves, including upregulation of genes characteristic of biotic stress responses and downregulation of genes associated with the function and translation of chloroplasts. The majority of plant genes that were differentially regulated during the symptomatic stage returned to their basal levels after recovery. Thus, the extent of changes in the plant transcriptome was correlated with symptom intensity. However, we also identified genes that remained upregulated after the symptomatic stage or that were specifically induced at the symptom recovery stage. The list of genes that were upregulated at the symptom recovery stage was diverse and included several types of cysteine-rich antimicrobial peptides, notably two defensin-like genes that were specifically upregulated in recovered leaves, as confirmed by droplet-digital PCR. Several plant miRNAs were also differentially expressed in ToRSV-infected plants. Notably, miR391 was upregulated at both stages of infection, whereas miR530 and miR1919 were specifically upregulated during the symptomatic stage of infection. Several predicted miRNA targets were differentially regulated in our dataset, including new targets as well as previously validated targets (e.g., zinc finger A20/AN1 domain-containing stress-associated protein 1, a known target of miR530).

by the Minister of Agriculture and Agri-Food, 2024. This material is licensed under the terms of the Canada Open Government Licence, which permits unrestricted use, distribution, and reproduction in any medium, provided you acknowledge the source of the Information or link to any attribution statement specified in the license. To view this licence, visit https://open.canada.ca/en/open-government-licence-canada.

**Data availability statement:** The dataset has been deposited in NCBI Bioproject PRJNA1253310 https://www.ncbi.nlm.nih.gov/bioproject/?term=PRJNA1253310.

**Funding:** Agriculture and Agri-Food Canada core funding.

**Competing interests:** The authors have declared that no competing interest exist. The funders had no role in study design, data collection and analysis, decision to publish, or preparation of the manuscript.

Many of the miRNA predicted targets were related to plant defense responses and may contribute to symptom induction and/or symptom recovery.

---

## Introduction

Virus infection induces multiple changes in the host cellular metabolism to create a suitable environment for viral replication, translation and movement. The plant responds by activating defence mechanisms to control virus multiplication and minimize the adverse effects of viral nucleic acids and proteins. These multi-level plant-virus interactions can result in symptom development, the severity of which is determined by the genetics of the host, the virulence of the infecting virus and environmental factors [1–4]. Symptom recovery is one of the possible outcomes of compatible plant-virus interactions, where young emerging leaves are asymptomatic after an initial systemic symptomatic stage [5]. Symptom recovery was first reported from tobacco plants infected with tobacco ringspot virus (genus *Nepovirus*, family *Secoviridae*) [6].

Symptom recovery has been associated with the induction of plant antiviral RNA silencing mechanisms [5]. RNA silencing is triggered by the presence of double-stranded RNAs that originate during viral replication or that are formed on viral genomic RNAs due to foldback regions [7,8]. These double-stranded RNAs are sliced by DICER-like proteins into 21–24 nucleotide fragments, termed virus-derived small interfering RNAs (vsiRNAs) [7,8]. Plants also encode endogenous small RNAs, for example microRNAs (miRNAs) that regulate the expression of plant genes, including genes involved in the response to biotic stresses [7,9,10]. One strand of the small RNA is loaded onto ARGONAUTE (AGO) to form RNA-induced Silencing Complex (RISC) or RNA-induced Transcriptional Silencing Complex (RITS) [7,8]. RISC complex targets single-stranded RNAs based on complementarity with the loaded small RNA, resulting in post-transcriptional gene silencing (PTGS) via cleavage or translational repression of target RNA [7,8]. RITS complex represses DNA transcription via DNA methylation, resulting in transcriptional gene silencing (TGS) [7,8]. Most plant viruses encode suppressors of RNA silencing (VSR) that counteract RNA silencing at various steps [7,11].

Common as well as unique sets of defence responses are induced in response to each specific host-virus combination [5]. Recovery from geminivirus infection is associated with the induction of both TGS and PTGS, resulting in a rapid decrease in the accumulation of viral genomic DNA and transcripts [12]. Viral RNA concentration is also often reduced after symptom recovery from infection with positive-sense RNA viruses, including tobacco rattle virus (TRV, genus *Tobravirus*) [13], cucumber mosaic virus (CMV, genus *Cucumovirus*) [14] and deficient mutants of tomato bushy stunt virus (TBSV, genus *Tombusvirus*) that lack the p19 VSR [15]. This reduction in viral RNA concentration has been attributed to the induction of PTGS [16]. However, symptom recovery is not always accompanied with reduced accumulation of viral RNAs as has been shown for some nepoviruses [17–20].

Analysis of the transcriptome of virus-infected plants provides a snapshot of the relative abundance of viral and host RNAs during the course of infection and can provide important insights into the regulation of host gene expression at specific stages of infection [21,22]. Most published analyses have focused on the transcriptome of symptomatic plants. Only a few studies have examined the transcriptome of infected plants at the recovery stage. In CMV-infected plants, recovery is characterized by having a much lower number of differentially expressed genes (DEGs) compared with the symptomatic stage, but also by the specific enrichment of genes involved in plant-pathogen interactions [23]. Similarly, the extent of transcriptomic changes in *Nicotiana benthamiana* plants infected with grapevine fanleaf virus (GFLV, a nepovirus) is correlated with symptom intensity, with much fewer DEGs associated with the recovery stage of infection [17]. In contrast, the transcriptome of plants infected with pepper golden mosaic virus (a geminivirus) does not show major differences during the symptomatic and recovered stages of infection [24]. However, a study on a distinct geminivirus (South African cassava mosaic virus) identified several resistance-like genes that are induced during the recovery stage [25].

Tomato ringspot virus (ToRSV, species *Nepovirus lycopersici*) belongs to the genus *Nepovirus*, family *Secoviridae* [26,27]. ToRSV has a wide host range and causes serious diseases in fruit trees, small fruits and grapevine [28,29]. The genome consists of two positive-strand RNAs, each encoding one large polyprotein. RNA1 encodes replication proteins and RNA2 encodes the movement protein (MP) and coat protein (CP). RNA1 and RNA2 share long stretches of nucleotides that are identical or nearly identical in both their 5' region and 3' untranslated region (UTR). Several ToRSV isolates have been characterized that vary in their virulence [30,31]. ToRSV-Rasp1 is a severe raspberry isolate that triggers an hypersensitive reaction (HR)-like response in *N. benthamiana* [20]. Necrotic rings first develop on the inoculated leaves followed by the appearance of vein-clearing symptoms on upper non-inoculated leaves. These symptomatic leaves are characterized by evidence of cell death, increased production of reactive oxygen species and induction of PR1a (pathogenesis-related protein) expression [18,20]. Down-regulation of ribulose bisphosphate carboxylase, a photosynthesis-associated gene, was also noted during the symptomatic stage of infection [18,30]. Infected plants grown at low temperature (21°C) continue to exhibit systemic necrosis leading to plant death. In contrast, plants grown at 27°C recover from infection and later produce young leaves devoid of symptoms [18,30]. We have previously shown that symptom recovery of *N. benthamiana* from ToRSV-Rasp1 is accompanied by sequence-specific RNA silencing but not by a concurrent decrease in the accumulation of the viral RNA2 [18,20]. We also reported that the translation of ToRSV-Rasp1 RNA2 is repressed in recovered leaves. Silencing of *NbAGO1* prevents symptom recovery and relieves the RNA2 translation repression [18]. A transient spike in the accumulation of *NbAGO2* transcripts is observed at early stages of infection (3–5 days post-inoculation, dpi) at both 21 and 27°C, but a concurrent spike in the accumulation of AGO2 protein is only observed in plants that later recovered from infection (i.e., at 27°C but not at 21°C) suggesting that post-transcriptional regulation of NbAGO2 expression occurred [30]. The accumulation of ToRSV RNAs and CP is enhanced in an *ago2* mutant, but symptom recovery is not prevented.

A previous microarray study revealed induction of a variety of stress-related genes (e.g., PR proteins, heat shock proteins, chaperones) and down-regulation of photosynthesis-associated genes during the symptomatic stage in ToRSV-infected *N. benthamiana* plants [32]. However, transcriptomic changes associated with symptom recovery have not been characterized in ToRSV-infected plants. In this study, we sequenced total RNAs and small RNAs extracted from ToRSV-Rasp1-inoculated *N. benthamiana* plants during the symptomatic and recovery stages of infection. We show extensive reprogramming of the transcriptome during the symptomatic stage of infection, particularly down-regulation of chloroplast functions and induction of genes related to stress response and protein folding. The recovery stage was associated with much fewer transcriptomic changes, with most genes returning to their basal levels from their differentially regulated state observed during the symptomatic stage. However, we also note the specific late induction of a few defense-related genes after recovery, notably defensins. Finally, we identify several miRNAs that are induced during ToRSV infection and show differential regulation of some of their predicted target genes.

## Results and discussion

### Experimental setup and initial examination of the transcriptomic data

ToRSV-Rasp1 was used to inoculate plants, which were grown at 27°C in a controlled growth chamber. Symptoms induced by ToRSV-Rasp1 in *N. benthamiana* have been described in details [18,20]. At 27°C, plants typically exhibit necrotic rings on inoculated leaves and vein clearing symptoms on young upper systemically infected leaves at three days post-inoculation (3 dpi) (Fig 1A and 1B). By 10 dpi, symptom recovery is established and young upper leaves are asymptomatic (Fig 1C). We collected samples from young upper leaves at these two time points. Each biological repeat consisted of a separate infection experiment using a separate batch of plants. To account for plant-to-plant variation, leaf samples were collected from 3–5 plants (2 upper non-inoculated leaves per plant) and pooled for total RNA extraction. For each biological repeat, half of the plants were mock-inoculated with inoculation buffer and samples were collected at the same time points using equivalent leaves (leaves of similar size, age and position, Fig 1).

Due to budget limitations, the first and second biological repeats were conducted in 2015/16 and the third in 2018. Total RNA sequencing of all repeats was carried out using 75 bp paired-end sequencing. We received 28 million to 35 million trimmed reads per sample for the first two repeats, while the third repeat had nearly five times the number of reads (137–219 million). For small RNA sequencing, the first two repeats used 75 bp paired-end sequencing, which produced 16–22 million trimmed reads, and the third repeat used 75 bp single-end sequencing, which produced two to four million trimmed reads per sample (Table 1). The dataset was deposited in NCBI (National Center for Biotechnology Information): BioProject PRJNA1253310.

Principal component analysis (PCA) of total and small RNA sequencing data shows that the third repeat differs from the first two repeats and clusters separately (Fig 2A and 2B). Although we did not note major differences in symptom development between the three repeats, we have previously observed that the specific timing and accumulation of transcripts can vary from one infection experiment to another, as documented for the transient spike in levels of AGO2 transcripts [30]. In addition, environmental conditions (light, humidity) may have also contributed to the differences between the repeats conducted in 2016 and 2018. However, the PCA analysis also revealed that mock-inoculated (3 and 10 dpi) and recovered (10 dpi) leaf samples clustered together, separately from the symptomatic (3 dpi) leaf samples for all repeats (Fig 2A and 2B). Thus, the extent of transcriptomic changes was correlated with the presence and intensity of visible symptoms, as previously reported for the GFLV-*N. benthamiana* interaction [17].

To gain additional insights into replicate variability, we also analyzed the dataset by hierarchical clustering using Pearson correlation and a filtering setting for the top 1000 genes. Similar to the PCA plots, the results show that repeat 3 differs from repeats 1 and 2 (S1 Fig). However, the results also reveal the separation of symptomatic samples from mock or recovered samples. Indeed, close examination of the heat map highlight a large number of DEGs that are shared amongst all three symptomatic samples, but are not differentially regulated in mock or recovered samples (S1 Fig).

### Viral RNA and vsiRNA dynamics during the symptomatic and recovery stages of infection

The ToRSV-Rasp1 isolate has been maintained in *N. benthamiana* plants in our lab by serial passaging. The two genomic RNAs were sequenced in 2014 (NCBI accession number: KM083894, KM083895) [31]. Serial passaging of the virus may result in genetic drift, which we tested by assembling the RNA1 and RNA2 sequences from each biological repeat using both small RNA and total RNA reads and comparing the assembled sequences with the ToRSV-Rasp1 reference genome. A few single-nucleotide point mutations were observed (Table 2, S1 Text and S2 Text). Four mutations occurred before the start of experiments, as these are present in the viral genomic sequence assembled from the first repeat and are maintained in subsequent repeats. One mutation in the 3' UTR ($G_{6907}$ -> $A_{6907}$), in a region unique to RNA1, was first detected in the second repeat and was also observed in the third repeat. Three additional mutations occurred between the second and third repeats. Two of these mutations are located within the polyprotein coding regions but are synonymous. The

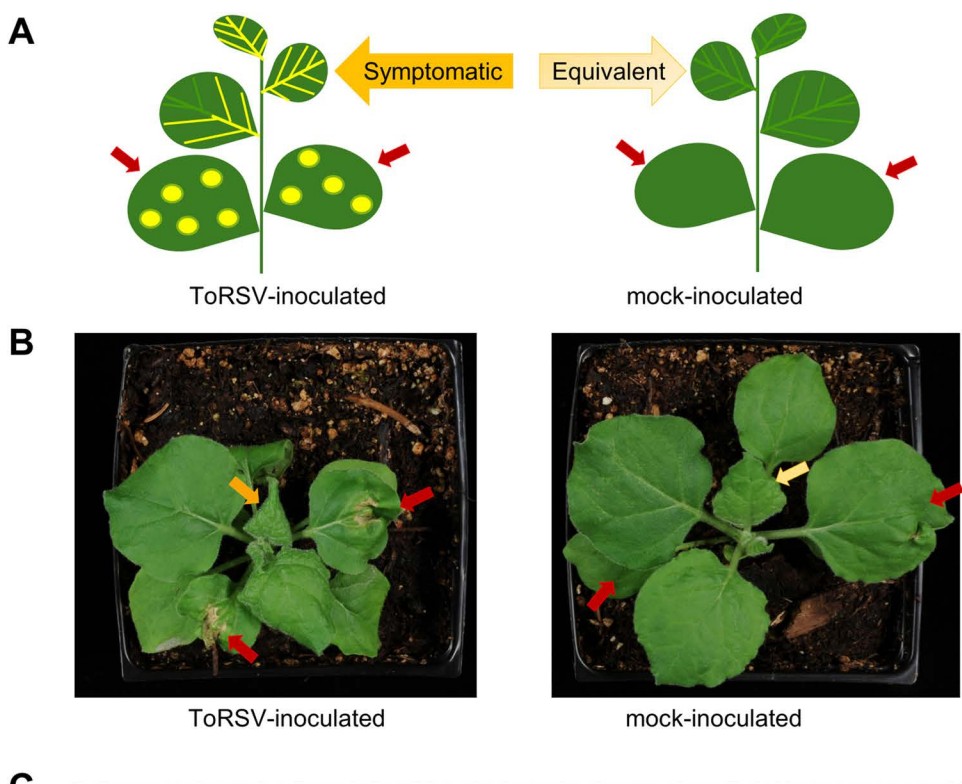

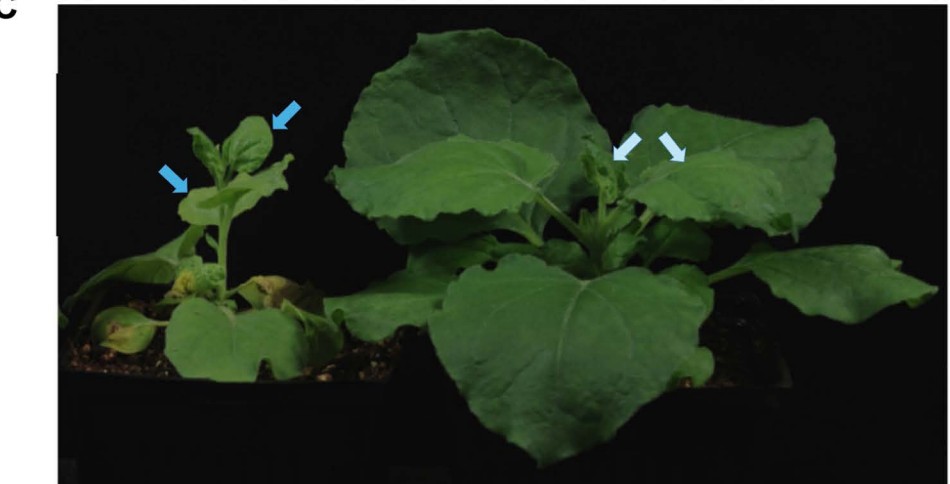

**Fig 1. Typical ToRSV-induced symptoms and sampling method. (A)** Schematic diagram illustrating the sampling method. At 3 dpi, ToRSV-inoculated *N. benthamiana* plants exhibit localized necrotic rings on the lower inoculated leaves (red arrow), and vein-clearing symptoms on the upper systemically infected leaves (orange arrow). Samples were taken from upper systemically infected symptomatic leaves (orange arrow) or equivalent leaves (light-yellow arrow) from mock-inoculated plants at 3 dpi. **(B)** Representative pictures of a ToRSV-infected plant and a mock-inoculated plant at 3 dpi. Leaves are marked as in (A), with typical sampled leaves shown by the orange and light-yellow arrows. **(C)** Representative picture of a ToRSV-infected plant at the recovery stage alongside a mock-inoculated plant (picture taken at 11 dpi). Typical sampled leaves are marked with the blue arrows. Although ToRSV-infected plants are stunted compared to mock-inoculated plants, the upper recovered leaves are asymptomatic.

**Table 1. Number of trimmed reads obtained for total RNA and small RNA sequencing for each sample.**

| Samples | Inoculation | Time of collection (dpi) | Biological Repeat | Reads (Total RNA) | Reads (small RNA) |
|---|---|---|---|---|---|
| Symptomatic | ToRSV-Rasp1 | 3 | 1 | 28,208,394 | 17,582,770 |
| Symptomatic | ToRSV-Rasp1 | 3 | 2 | 32,120,198 | 20,069,875 |
| Symptomatic | ToRSV-Rasp1 | 3 | 3 | 137,935,812 | 3,051,151 |
| Recovered | ToRSV-Rasp1 | 10 | 1 | 30,250,006 | 17,233,648 |
| Recovered | ToRSV-Rasp1 | 10 | 2 | 35,409,562 | 22,099,790 |
| Recovered | ToRSV-Rasp1 | 10 | 3 | 185,565,352 | 4,278,884 |
| Mock 3 dpi | Mock | 3 | 1 | 32,019,302 | 18,395,507 |
| Mock 3 dpi | Mock | 3 | 2 | 27,339,784 | 16,019,315 |
| Mock 3 dpi | Mock | 3 | 3 | 219,300,854 | 2,833,616 |
| Mock 10 dpi | Mock | 10 | 1 | 30,520,434 | 19,188,185 |
| Mock 10 dpi | Mock | 10 | 2 | 33,907,386 | 21,427,346 |
| Mock 10 dpi | Mock | 10 | 3 | 171,761,566 | 4,476,402 |

third mutation located in the RNA1 3' UTR was distinct between RNA1 and RNA2 in repeats 1 and 2, but became identical in both RNAs in repeat 3. None of the 3' UTR mutations are in region 3 of the 3' UTR, which was previously shown to facilitate the cap-independent translation of RNA2 [33]. Although the four mutations detected during the course of the experiments (i.e., mutations first detected in repeats 2 or 3) may affect the stability, translation and/or replication of the viral RNAs, they would not change the amino acid sequence of the encoded viral proteins.

Next, we examined the accumulation of RNA1 and RNA2 at the two time points for each repeat. Trimmed reads were mapped to the viral RNAs to give the relative number of reads for each RNA (reads per kilobase per million reads or RPKM) (Table 3). The viral RNAs were detected at similar levels in symptomatic and recovered leaves. This result was consistent with our previous observations that symptom recovery is not associated with a reduction in viral RNA2 concentration [18,20]. The ratio of RNA2 to RNA1 varied from 0.9 to 2.8 and was confirmed by ddPCR validation (see below). This ratio is much lower than the reported RNA2:RNA1 ratio of 10–100 fold for *N. benthamiana* plants infected with another nepovirus (GFLV) [17].

Small RNA reads were mapped to the viral genome to visualize the distribution of vsiRNAs. These were distributed throughout the genome, although some hotspots were identified, notably in the coding regions for the putative nucleoside triphosphate binding protein (NTB) in RNA1, and for the MP and CP in RNA2 as well as the 3' UTRs of the RNAs (Fig 3). These hotspots indicate preferential target sites for the host silencing machinery, which could be used for developing virus resistance [34]. Comparing the distribution of vsiRNAs to the 3' UTRs of RNA1 and RNA2 show patterns that are generally similar, except for a few specific hotspots. These results can be explained by our previous observation that the 3' UTRs of ToRSV-Rasp1 RNA1 and RNA2 share an overall 81% nucleotide sequence identity, which is unevenly distributed along the 3' UTRs with stretches of near-complete sequence identity interspersed by more divergent regions [31,33]. Some minor changes in the distribution of the vsiRNAs were also noted between the symptomatic and recovery stages of infection. In a previous study, we had divided the ToRSV genome into 15 fragments of 1000 nts and evaluated the concentration of vsiRNAs mapping to each fragment using hybridization techniques [20]. Although the plants were grown under different environmental conditions (greenhouse conditions with fluctuating temperature), similar hot spots of vsiRNAs were previously identified on ToRSV RNA2, notably in the MP and CP coding regions [20]. This is evidenced by the mapping of vsiRNA reads to the 1000 nts genome fragments used in the previous study, with fragments To12, To13 and To14 showing a high number of mapped vsiRNA reads at both the 3 dpi and 10 dpi time points (S2 Fig).

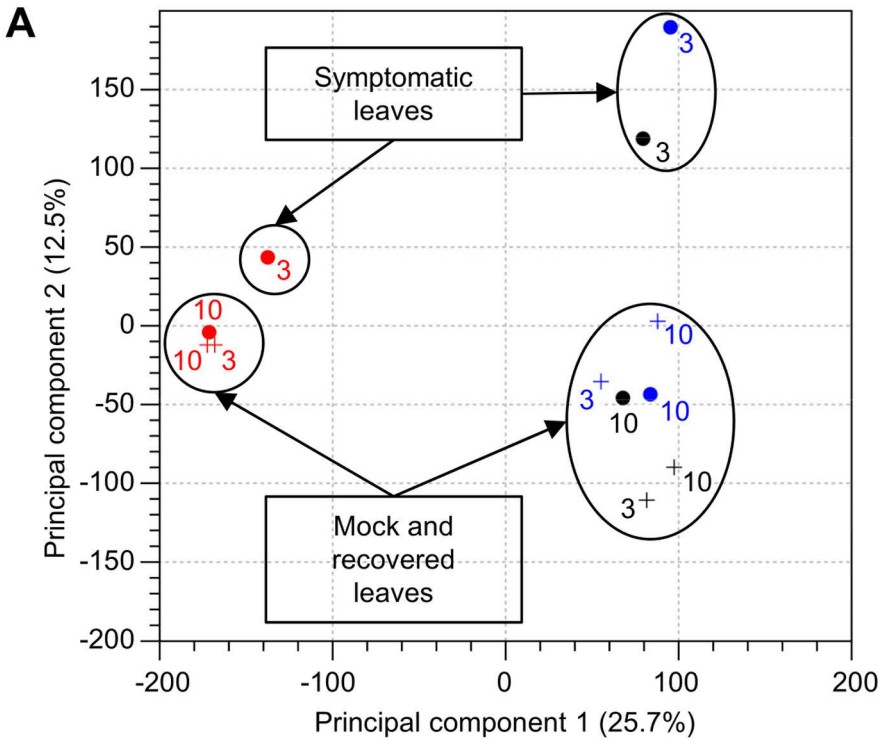

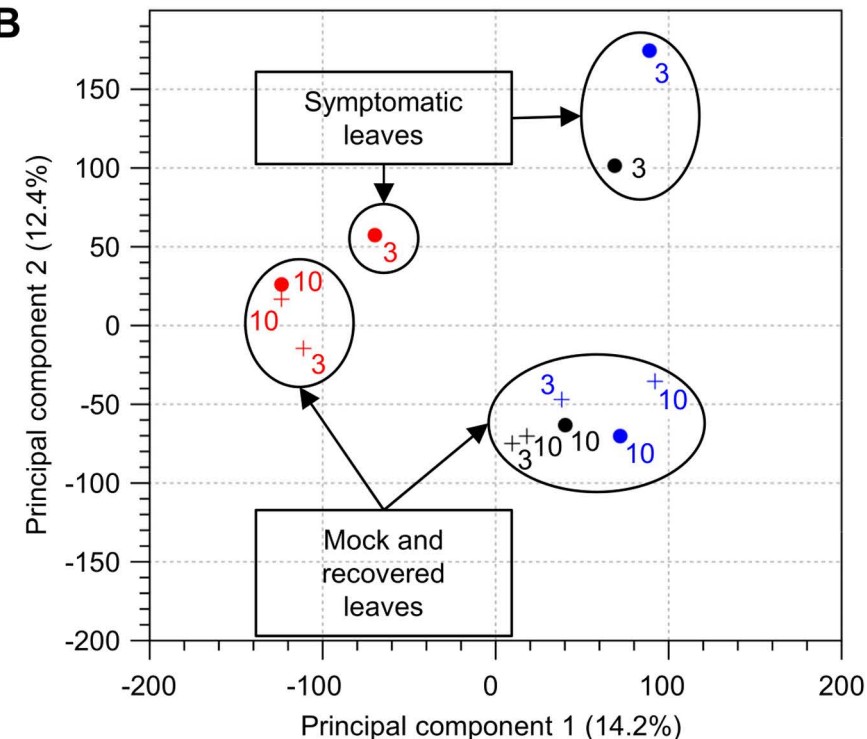

**Fig 2. Principal component analysis plot of (A) total RNA and (B) small RNA sequencing data.** Sequences from repeats 1, 2 and 3 are color coded in black, blue and red, respectively. The numbers beside each data point indicates the number of days post-inoculation (3 dpi or 10 dpi) for mock-inoculated samples (+) or ToRSV-inoculated samples (•).

**Table 2. Nucleotide changes in the viral genomic RNAs compared to the reference genomic sequence of ToRSV-Rasp1 (NCBI accession number: KM083894, KM083895). Single nucleotide mutations and their corresponding location in the viral reference genome are denoted by letters with subscript numbers. Single-nucleotide mutations in the polyprotein coding region are shown with the resulting change in amino acids. Mutations that occurred during this experiment (i.e., after Repeat 1) are marked with an asterisk (*).**

| Mutation | Changes | RNA | Repeat 1 | Repeat 2 | Repeat 3 |
|---|---|---|---|---|---|
| $UAU_{4835}$ -> $UAC_{4835}$ | CDS (Tyr -> Tyr) | RNA1 | Yes | Yes | Yes |
| $AA_{5812}A$ -> $AG_{5812}A$ | CDS (Lys -> Arg) | RNA1 | Yes | Yes | Yes |
| $UGU_{2269}$ -> $UGC_{2269}$ | CDS (Cys -> Cys) | RNA2 | Yes | Yes | Yes |
| $GG_{2994}U$ -> $GA_{2994}U$ | CDS (Gly -> Asp) | RNA2 | Yes | Yes | Yes |
| $G_{6907}$ -> $A_{6907}$ | 3' UTR | RNA1 | No | Yes* | Yes |
| $ACU_{3695}$ -> $ACC_{3695}$ | CDS (Thr -> Thr) | RNA1 | No | No | Yes* |
| $G_{7384}$ -> $A_{7384}$ | 3' UTR | RNA1 | No | No | Yes* |
| $UGC_{1792}$ -> $UGU_{1792}$ | CDS (Cys -> Cys) | RNA2 | No | No | Yes* |

**Table 3. Differential accumulation of viral RNAs in symptomatic and recovered leaves. Concentration of each viral RNA was calculated from the total RNA sequencing data and expressed in rpkm (reads per kilobase per million reads).**

| Repeat | Infection status | RNA1 (rpkm) | RNA2 (rpkm) | Ratio (RNA2:RNA1) |
|---|---|---|---|---|
| Repeat 1 | Symptomatic | 63,909 | 62,728 | 0.98 |
| | Recovered | 45,629 | 82,606 | 1.81 |
| Repeat 2 | Symptomatic | 65,583 | 60,907 | 0.93 |
| | Recovered | 46,071 | 82,125 | 1.78 |
| Repeat 3 | Symptomatic | 43,930 | 84,453 | 1.92 |
| | Recovered | 33,919 | 95,339 | 2.81 |

## *N. benthamiana* transcriptomic response at various stages of ToRSV-Rasp1 infection

To understand the transcriptomic response of *N. benthamiana* during virus infection, differentially expressed genes (DEGs) were identified in pairwise comparison of transcriptomes between symptomatic and equivalent mock-inoculated leaves (3 dpi), and between recovered and equivalent mock-inoculated leaves (10 dpi) (Fig 4A and 4C; S1 File and S2 File) (absolute fold change ≥2 or ≤0.5 corresponding to log2fold change ≥1 or ≤−1, p value ≤0.05 and max group mean ≥5.0). The number of DEGs was much larger in the symptomatic vs mock comparison (920 DEGs) than in the recovered vs. mock comparison (177 DEGs). These results are consistent with previous studies that have shown a correlation between the number of DEGs and symptom severity in plants infected by various positive-strand RNA viruses [17,23,32,35]. We also compared the transcriptome of recovered leaves to that of symptomatic ones (Fig 4A and 4C; S3 File). To confirm that transcriptomic changes between the recovered and symptomatic stages were not simply due to the age of the plants, we also compared the transcriptomes of mock-inoculated leaves at the two time points (S4 File). In the recovered vs symptomatic comparison 858 DEGs were observed, while only 18 DEGs were observed in the comparison of the mock controls (10 dpi vs. 3 dpi). These two comparisons shared only two DEGs.

Of the 920 DEGs observed in the symptomatic vs. mock comparison, 537 were upregulated, and 383 were downregulated (S1 File). Most of these DEGs returned to their basal levels in recovered leaves and in fact were differentially regulated in opposite directions when comparing recovered leaves to symptomatic leaves (S1 File, Fig 4C). In recovered leaves, 177 DEGs were upregulated compared to the corresponding mock controls (S2 File). We did not find any downregulated DEGs in this comparison.

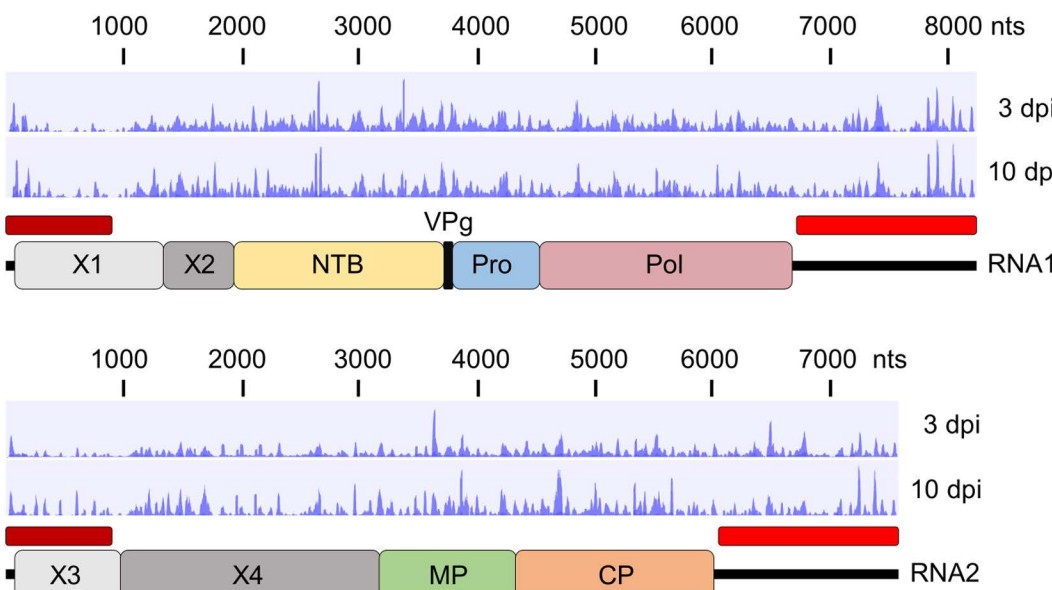

**Fig 3. Mapping of virus-derived small RNAs to the viral genome.** The coding regions for protein domains in the RNA1 and RNA2 polyproteins are shown with the boxes. RNA1 and RNA2 share 81% nucleotide sequence identity on their 3' UTRs (red bar). The 5' UTRs and the 5' ends of the X1 and X3 coding regions of RNA1 and RNA2, respectively, also share extensive nucleotide sequence identity (dark red bar). NTB: putative nucleoside triphosphate binding protein, VPg: virus genome-linked protein, Pro: protease, Pol: RNA-dependent RNA polymerase, MP: movement protein, CP: coat protein. The functions of proteins X1, X2, X3 and X4 are not known.

## Identification of genes that are differentially upregulated in symptomatic leaves

First, we conducted singular enrichment analysis (SEA) using separate lists for up and downregulated genes. Out of 537 upregulated genes in symptomatic leaves, 338 were annotated in the query list and compared to a background of 31449 annotated genes. This analysis revealed two enriched gene ontology terms (GOs) for the biological processes category and 10 GOs for the molecular function category (Fig 5, S5 File). GO terms 'protein folding', 'unfolded protein binding' and 'chaperone binding' were the most significantly enriched. Other enriched terms were related to 'ion binding', more specifically 'metal ion binding' (including a number of zinc-finger and ring-finger domain proteins) and 'calcium ion binding', as well as 'divalent inorganic cation transmembrane transporter activity' and 'protein serine/threonine phosphatase activity'. These results are consistent with a previous microarray study that showed induction of biotic stress responses in symptomatic ToRSV-infected *N. benthamiana* plants, including WRKY transcription factors, heat shock proteins and putative disease resistance proteins, as well as genes related to metal ion binding, the ubiquitin/proteasome system (UPS) and membrane intracellular transport [32].

Examination of the list of upregulated genes confirmed the induction of many chaperones, including heat-shock proteins (such as HSP70) and peptidyl-prolyl cis-trans isomerase D (PPctI-D), which were amongst the top 10 upregulated DEGs (S1 File). Most of these genes returned to basal level of expression after symptom recovery. Heat shock proteins are often induced during virus infection and are part of the plant defence response [36]. Peptidyl-prolyl cis/trans isomerases are a unique family of molecular chaperones that regulate protein folding at proline residues. They are also called cyclophilins [37]. Cyclophilins have been shown to impact the replication of a tombusvirus, a begomovirus and a closterovirus [38–40]. Upregulation of HSP70 and cyclophilin was confirmed in our droplet-digital PCR assays (see below).

Several putative disease resistance proteins were upregulated, including an NB-ARC domain-containing disease resistance protein (Niben101Scf04563g01014), which was amongst the top 10 upregulated genes in both symptomatic and

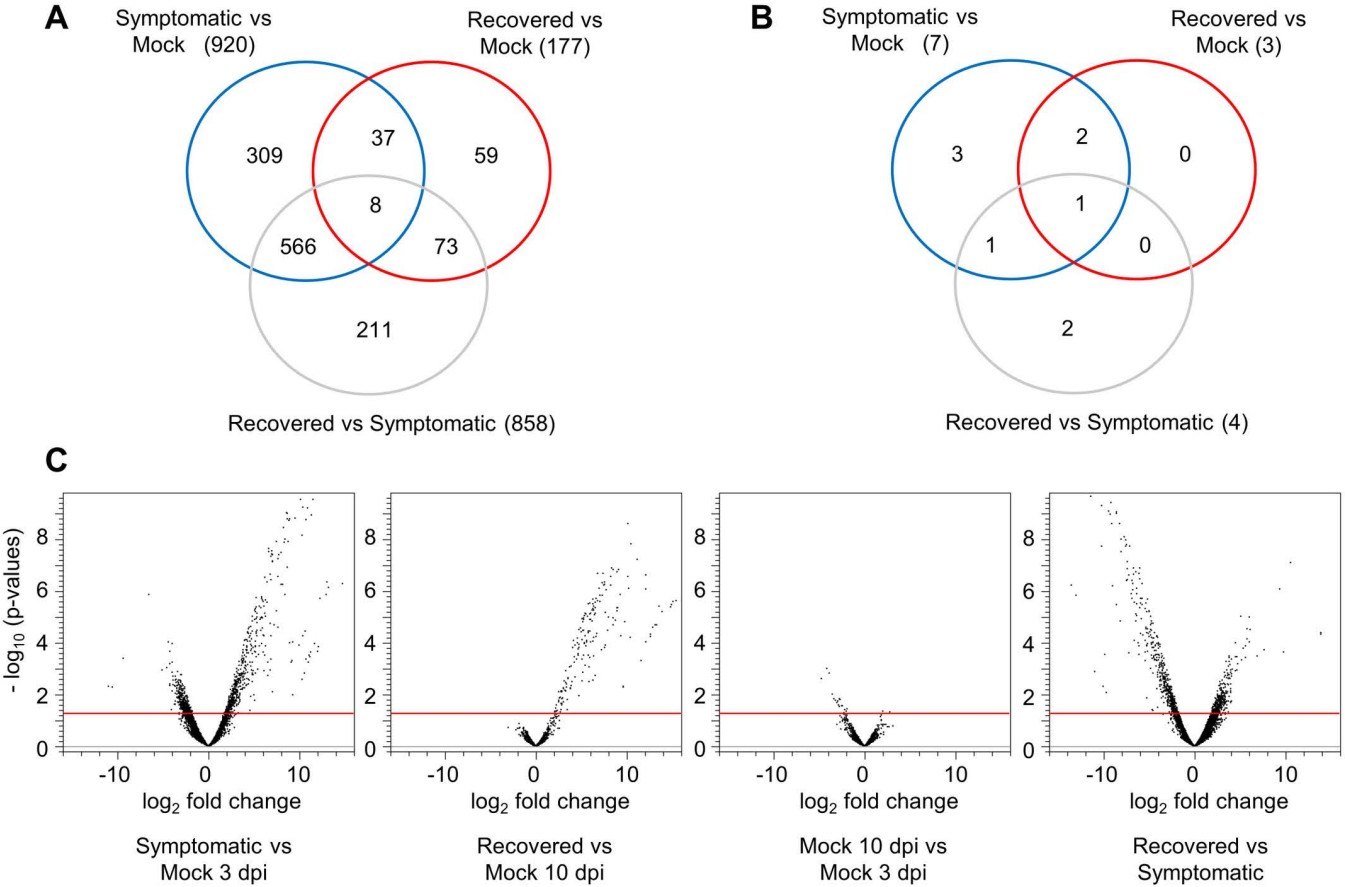

**Fig 4. Differential expression of plant genes and miRNAs during ToRSV-Rasp1 infection. (A)** Venn Diagram showing differentially expressed genes (1,263 DEGs). **(B)** Venn diagram showing differentially expressed miRNA (9 miRNAs). Only DEGs or miRNAs with absolute fold changes ≤ 0.5 or ≥ 2, p-value ≤ 0.05 and maximum group mean ≥ 5 are included in the analyses. **(C)** Volcano plot displaying DEGs in four pairwise comparisons. RNA sequencing reads mapped to *N. benthamiana* were analyzed and pairwise comparisons were made as indicated below each panel. DEGs with max group mean value < 5 were excluded from the analysis. The red bar is set at –log10 (p-value) of 1.3 (corresponding to a p-value of 0.05). Dots above the bar are differentially expressed in a significant manner. All dots above the bars also corresponded to DEGs with absolute fold changes ≤ 0.5 or ≥ 2.

recovered leaves (S1 File and S2 File). Resistance proteins are associated with pathogen recognition and trigger plant defence responses [41]. Various transcription factors were also induced in symptomatic leaves, including several WRKY factors, which are known to be involved in plant responses to biotic and abiotic stresses [42,43].

Given the anticipated role of RNA silencing in symptom recovery, we examined the expression of silencing-related genes. We have previously reported a transient spike of AGO2 expression at early stages of infection [30]. This was confirmed in our transcriptomic dataset (see S1 File, Niben101Scf05245g01007, transcript induced in symptomatic leaves but not in recovered leaves). Increased accumulation of ToRSV RNAs and CP was previously observed in a *N. benthamiana* mutant defective in AGO2, but symptom recovery was not prevented in this mutant [30]. We did not observe differential regulation of other RNA silencing-related genes in our dataset.

As noted in a previous study [32], several components of the UPS were induced, including proteasome subunits, ubiquitin-protein ligases, ubiquitin fusion-degradation protein and ubiquitin conjugating enzymes, all of which returned to basal levels in recovered leaves with the exception of one ubiquitin-protein ligase (Niben101Scf11773g01006), which still showed increased expression after symptom recovery. The UPS plays both antiviral and pro-viral roles in infected

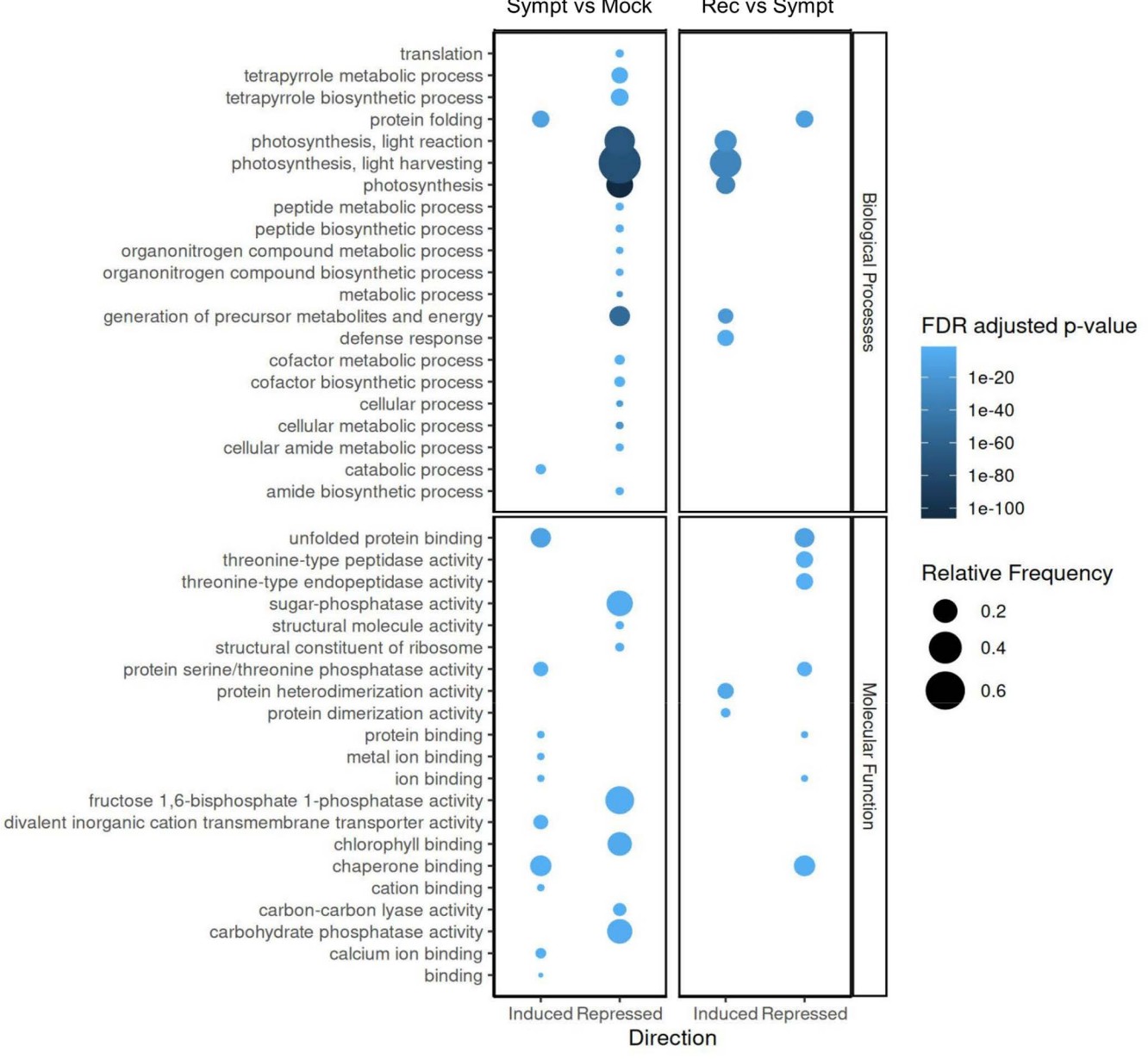

**Fig 5. SEA analysis for DEGs that are induced or repressed in the symptomatic vs mock (Sympt vs Mock) or recovered vs symptomatic (Rec vs Sympt) comparisons.** Circle size and colors indicate level of enrichment and FDR adjusted P values, respectively.

plants, by directing the degradation or modification of viral proteins and of plant proteins involved in various defense response pathways [44,45]. We have previously shown that the UPS contributes to the destabilization of AGO2 in ToRSV-infected plants [30]. The UPS may also regulate the accumulation of viral proteins, although this has not been rigorously tested.

Proteins with signature domains for protein-protein or protein-DNA interactions were identified that could play multiple roles in virus infection. At least five ankyrin repeat-containing proteins were upregulated in symptomatic leaves,

including one which was amongst the top 10 upregulated genes in both symptomatic and recovered leaves (S1 File and S2 File, Niben101Scf01328g01020). Ankyrin repeat-containing proteins facilitate protein-protein interactions and are found in a variety of proteins, including membrane channel proteins, enzymes, toxins, transcription factors and signal transduction proteins [46]. Several zinc-finger domain-containing proteins were also identified, notably zinc finger FYVE domain-containing protein 21 (Niben101Scf09812g01018), a component of the endosomal sorting complex required for transport (ESCRT) machinery, which is implicated in abiotic stress response and the regulation of multivesicular endosomal formation and lipid droplet turnover [47–49] and zinc finger protein 1 (Niben101Scf08719g00002), a C2H2-type zinc-finger protein previously shown to interact with histone genes in wheat [50]. Several C2H2-type zinc-finger proteins were previously found to be induced upon infection with cucumber mosaic virus and turnip mosaic virus [51].

## Identification of genes that are differentially downregulated in symptomatic leaves

SEA analysis of the 383 downregulated genes identified 260 genes annotated in the query list, yielding 52 significant GO terms: 18, 7 and 27 in the biological processes, molecular function and cellular component categories, respectively (Fig 5, S5 File). The most significantly enriched GO terms were related to photosynthesis and chloroplast function, notably 'photosynthesis', 'light harvesting', 'chlorophyll binding', 'fructose 1,6-biphosphate 1-phosphatase activity' (a key step of the Calvin cycle), as well as 'cellular component of the thylakoid'. We have previously shown downregulation of the rbcS gene (coding for the small subunit of RUBISCO) in symptomatic leaves but not in recovered leaves using Northern blots [18]. This is also observed in our transcriptomic dataset (S1 File, Niben101Scf04196g00005, Niben101Scf03015g06014 and Niben101Scf01991g04010), along with many other photosynthesis-related genes. Other significantly enriched GOs related to 'cellular metabolic process' and 'translation', including 'tetrapyrrole biosynthetic process', 'peptide biosynthetic process' and 'structural constituent of the ribosome'. A previous study already documented down-regulation of plastid genes and of ribosomal subunits (mostly from plastid ribosomes) in symptomatic leaves of ToRSV-infected *N. benthamiana* plants [32]. Examination of the genes related to the GO 'structural constituent of the ribosome' confirmed that the majority of the downregulated genes were components of plastid ribosomes including 50S ribosomal proteins PRSP5, PRSP6, L4, L5, L7/L12, L10, L13, L17, L18, L27, L28, L29 and L34 as well as 30S ribosomal proteins, S6, S9, S10, S17, S20 and S31. A single cytosolic ribosomal protein (60S ribosomal protein L31) was significantly repressed. The down-regulation of plastid functions (photosynthesis, translation) is probably related to the mosaic and vein-clearing symptoms observed during the symptomatic stage of infection. The chloroplast, where signaling molecules such as reactive oxygen species, salicylic acid and jasmonic acid are produced, has emerged as a key regulator of plant defense responses [52–55]. Thus, inhibition of chloroplast functions may also reflect viral counter-defense responses to mitigate the production of the plant defense-related signaling molecules listed above.

Six aquaporin-like protein-encoding genes were downregulated in the symptomatic vs mock comparison. Aquaporins are membrane-integrated proteins that serve as channels for transporting water, gas, nutrients and removing reactive oxygen species [56]. The expression of aquaporins is affected by stress responses and can also mediate defence responses by transporting signalling molecules across the membrane [56]. In addition, some aquaporins interact directly with viral proteins. For example, the *A. thaliana* tonoplast intrinsic proteins TIP1 and TIP2 interact with a component of the cucumber mosaic virus replicase and have been suggested to facilitate virus replication in association with the tonoplast [57]. ToRSV replication occurs in membranous complexes derived from the endoplasmic reticulum [58,59]. Further experimentation would be required to investigate the possible impact of aquaporin downregulation on the replication of ToRSV RNAs.

## Identification of genes that are differentially upregulated in recovered leaves

When the transcriptome of recovered leaves was compared to that of corresponding mock-inoculated control 177 DEGs were found to be significantly upregulated and of those, 103 were annotated (S2 File). SEA analysis did not reveal any enriched GO terms for this comparison, but manual inspection of the list revealed a variety of DEGs with diverse biological

functions. We also performed SEA analysis for the comparison of recovered leaves to symptomatic leaves. Because many differentially regulated genes observed in symptomatic leaves returned to their basal levels in recovered leaves, most of the GOs enriched in the recovered vs mock comparison were similar to those enriched in the symptomatic vs mock comparison (Fig 5, S5 File). For example, GOs that were associated with downregulated DEGs in the symptomatic vs mock comparison (e.g., GOs related to photosynthesis or chloroplast function) were enriched when analyzing DEGs that were upregulated in the recovered vs symptomatic comparison. One exception was GO:0006952 (defense response), which was not enriched in the symptomatic vs mock comparison but was associated with nine significantly upregulated DEGs in the recovered vs symptomatic comparison. Close analysis of these nine DEGs revealed that they all showed downregulation trends in symptomatic leaves (symptomatic vs mock comparison), although these were not always considered significant either because the max group mean value was below 5 or because the p-value was above 0.05 (DEGs highlighted in yellow in S3 File). Seven of the nine DEGs returned to basal levels in recovered leaves. Of the nine DEGs associated with GO:0006952 in the recovered vs symptomatic comparison, four were defensin genes. Interestingly, two defensin genes showed a marked upregulation in the recovered vs mock comparison (Niben101Scf06275g03010 and Niben101Scf06275g03011). As shown below, ddPCR validation confirmed the specific upregulation of Niben-101Scf06275g03010 at the recovery stage. Defensin-like proteins are small cysteine-rich proteins also referred to as pathogenesis-related protein 12 or PR12 [60,61]. Plants possess a substantial number of defensin genes that are crucial for plant growth and development. Some of these genes are also induced in response to abiotic and biotic stresses, including infection by insects, fungi, bacteria and viruses [62,63]. Many plant defensins have direct antifungal activities. They permeabilize the cell membrane of fungi and cause cell leakage [61]. Although defensins have been reported to be induced by infection with tobacco mosaic virus [64], their role in antiviral defense responses is not well understood. Human defensins have been shown to destabilize several viral proteins *in vitro*, including the protease of tobacco etch virus, which is closely related to the ToRSV protease [65]. However, the relevance of this observation has not been tested in virus-infected cells and it is not known whether plant defensins possess similar activities. Plant defensins are multifunctional. They interact with various signaling pathways implicated in plant organ development, and potentially also regulate other components of the plant defense responses [61,66].

Of the 177 upregulated DEGs in recovered leaves (compared to the equivalent mock control), 45 were also significantly upregulated in symptomatic leaves (symptomatic vs mock comparison, S2 File). Another 108 genes also showed trends of upregulation in the symptomatic vs mock comparison (i.e., log2fold change ≥ 1) although these trends were not significant based on our selected parameters. These observations suggest that very few genes are specifically upregulated at the symptom recovery stage. In fact, only 24 DEGs were found that did not show any trends of upregulation in symptomatic leaves compared to the equivalent mock control (highlighted in yellow in S2 File). The two defensin genes discussed above were among the top 5 upregulated genes in this category along with a thionin-like gene (Niben-101Scf09993g00001). Similar to defensins, thionins are small cysteine-rich proteins that are induced in response to pathogen infection and have antimicrobial activities [60,61]. They are also referred to as PR13. Expression of thionins is induced in plants infected with capsicum chlorosis virus, a tospovirus, and with chili leaf curl virus, a geminivirus [67,68]. Interestingly, thionin expression was higher and more sustained in plants that were resistant to chili leaf curl virus, compared to susceptible plants [68].

In the list of genes specifically upregulated in recovered leaves, a eukaryotic aspartyl protease family protein (Niben-101Scf07325g00027), also known as secreted aspartic protease 2 (Sap2), was the most upregulated. Sap2 is induced by various stresses, including salt stress [69] and *Pseudomonas syringae* infection [70]. Sap2 contains an N-terminal signal peptide and is secreted into the apoplast, where it cleaves the *P. syringae* MucD protein, resulting in inhibition of bacterial growth [70]. It is not known whether or not Sap2 could affect the stability of ToRSV proteins that are located in ER-associated vesicles (replication proteins) [58,59] or in cell wall-associated tubular structures (MP) [71]. Alternatively, Sap2 could be indirectly involved in defense responses. For example, other aspartyl proteases have been shown

to cleave PR-1b, releasing peptides that elicit innate and systemic acquired resistance [72]. In addition, processing of a BAG family molecular chaperone regulator 6 (BAG6) by an aspartyl protease (APCB1) triggers autophagy and resistance against the fungal pathogen *Botrytis cinerea* [73]. We note that BAG6 (Niben101Scf02175g06018) was amongst the top 20 upregulated genes in symptomatic leaves, but returned to basal levels in recovered leaves (S1 File).

We also considered genes that were already upregulated in symptomatic leaves and showed sustained or increased upregulation in recovered leaves. With the exclusion of Sap2, the top ten upregulated genes in the recovered vs mock comparison belonged to that category (S2 File). In fact, we note two genes that were amongst the top ten upregulated genes in both symptomatic and recovered leaves and are already discussed above: an ankyrin repeat-containing-like protein (Niben101Scf01328g01020) and a disease resistance protein (Niben101Scf04563g01014). The list of the top ten upregulated genes in recovered leaves also includes genes annotated as non-specific lipid-transfer protein-like protein (Niben101Scf05588g13020), NADH-quinone oxidoreductase subunit H (Niben101Scf02074g06079), germin-like protein 5-1 (Niben101Scf07566g00008) and kunitz trypsin inhibitor 1 (KTI1, Niben101Scf01971g01005).

Like defensins and thionins, non-specific lipid-transfer proteins (nsLTPs) are cysteine-rich antimicrobial peptides and are referred to as PR14 [60,61]. There are six types of nsLTPs in *Nicotiana* species [74]. The nsLTP induced in ToRSV-infected leaves has conserved motifs consistent with type 4 nsLTPs. Although type 4 nsLTPs have not been studied in detail in the context of plant-virus interactions, recent studies have highlighted contrasting roles for type 1 nsLTPs. Potato StLTP6 facilitates potato virus S infection by disrupting jasmonic acid signalling and antiviral RNA silencing [75]. *N. benthamiana* NbLTP1 was also identified as a pro-viral factor, enhancing infection by bamboo mosaic virus [76]. On the other hand, NbLTP1 promotes salicylic acid mediated defense responses against tobacco mosaic virus infection [77]. Interestingly, the cowpea cpLTP1 was also identified as an antiviral factor that interacts with and inhibits the protease of cowpea mosaic virus (another member of the family *Secoviridae*, although from the genus *Comovirus*) [78].

NADH-quinone oxidoreductase subunit H is part of the NADH dehydrogenase I (NDH) complex involved in aerobic respiration. Two distinct subunits of the NDH complex have been shown to act as antiviral factors against turnip mosaic virus and strawberry mottle virus (another member of the family *Secoviridae*, although belonging to the genus *Sadwavirus*) [79,80], although the impact of NADH-quinone oxidoreductase subunit H overexpression on plant-virus interactions is not known.

Germin and germin-like proteins are involved in peroxidase metabolism and in responses to biotic and abiotic stresses [81]. Interestingly, although germin-like protein 5-1 was upregulated in ToRSV-infected leaves, four distinct germin-like proteins (all from subfamily 3) were downregulated in symptomatic leaves.

KTI1 belongs to a large family of plant serine protease inhibitors that provide resistance to insect herbivores by inhibiting their digestive enzymes [82,83]. KTIs are also induced following treatment with fungal and bacterial elicitors [84] and following infection with positive and negative sense RNA viruses [85–87]. In *N. benthamiana*, a gene labelled as Kunitz peptidase inhibitor-like protein is induced by infection with either potato virus X or tobacco mosaic virus and acts as a proviral factor by increasing plasmodesmata permeability and facilitating viral cell-to-cell movement [88,89]. In contrast, a citrus miraculin-like protein 2 (another member of the KTI family) inhibits citrus tristeza virus infection by limiting virus movement and replication, in part by altering the endomembrane system and stimulating reactive oxygen species accumulation [90].

## Validation of selected DEGs by ddPCR

To validate the RNA seq results, we used droplet-digital PCR (ddPCR). We tested two biological repeats: the previously described Repeat 3 from 2018 and an additional repeat (termed Repeat 4 hereon), which was performed in 2016 immediately after Repeats 1 and 2, but was not sequenced. First, we examined the expression patterns of a few candidate reference genes. We previously used PP2A as a reference gene [30], which showed relatively stable expression across treatments in our ddPCR results. However, a slight induction of PP2A expression in symptomatic leaves (~ 2-fold) was

noted for both tested repeats (S3 Fig). We also examined the expression of actin using previously described primers [91]. Again, expression was relatively stable across treatments with a slight down-regulation observed in symptomatic leaves in both repeats (~2-fold). We decided to use both reference genes to normalize the expression of ToRSV RNAs and the selected candidate plant genes. Averaging the expression of both reference genes resulted in reasonably stable ratios of expression across the tested samples (S3 Fig).

We tested the accumulation of ToRSV RNA1 and RNA2. The ddPCR results confirmed the stable accumulation of viral RNAs in symptomatic and recovered leaves and an RNA2:RNA1 ratio between 1.4 and 2.3 (Fig 6). Next, we tested three genes that were characteristic of antiviral defense responses and were highly upregulated in symptomatic leaves in the NGS dataset (Fig 7). The ddPCR results showed that HSP70 (Niben101Scf02124g01003) and PPctl-D (Niben-101Scf01832g01020) were highly upregulated in symptomatic leaves in both biological repeats and returned to basal levels in recovered leaves. SAR8 (Niben101Scf0625g03013, annotated as mRNA inducible by salicylic acid or by TMV during systemic acquired resistance) was also highly upregulated in symptomatic leaves (15–50 fold) and remained upregulated in recovered leaves, although to a lower extent (2–15 fold). These results were consistent with the expression trends observed for these three genes in the transcriptomic dataset.

We also examined the expression of genes that were significantly upregulated in recovered leaves in the transcriptomic dataset. We tested a defensin-like gene (DFL, Niben101Scf06275g03010), which was upregulated in recovered leaves but not in symptomatic leaves as discussed above. The specific upregulation of this gene in recovered leaves was confirmed by ddPCR for both biological repeats (Fig 7). The eight other tested genes showed trends of upregulation to various degrees in symptomatic leaves in the transcriptomic dataset although not all were statistically significant. The ddPCR results indicated upregulation of AMD (Niben101Scf01847g05004, a chloroplastic amidotransferase similar to bacterial glutamyl-tRNA$^{Gln}$ amidotransferase) in recovered leaves but not in symptomatic leaves for both biological repeats. Six other genes did not show differential regulation in recovered leaves by ddPCR, although they were upregulated in symptomatic leaves in at least one of the two biological repeats. These included MORN3 (Niben101Scf12277g02009, MORN-motif repeat protein similar to AT1G21920.1 MORN3 or MRF1 protein), AP-4 (Niben101Scf08394g00001, transcription factor AP-4), SP1 (Niben101Scf01487g04011, signal peptidase 1), TIM (Niben-101Scf27837g00005, mitochondrial import inner membrane translocase subunit Tim17/Tim22/Tim23 family) and FREE (Niben101Scf09812g01018, zinc finger FYVE domain-containing protein 21, already discussed above). The last tested

**A**

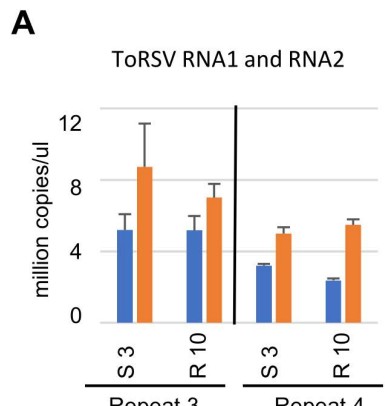

**B**

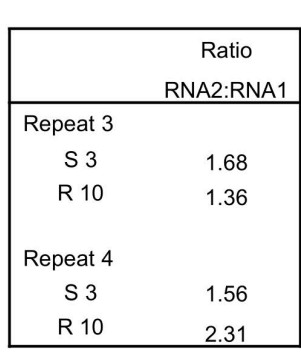

| | Ratio RNA2:RNA1 |
|---|---|
| Repeat 3 | |
| S 3 | 1.68 |
| R 10 | 1.36 |
| | |
| Repeat 4 | |
| S 3 | 1.56 |
| R 10 | 2.31 |

**Fig 6. Quantification of ToRSV RNA1 (blue) and RNA2 (orange) in total RNA samples extracted from symptomatic leaves at 3 dpi (S 3) or recovered leaves at 10 dpi (R 10). (A)** Total RNA samples were diluted 1:2,000 and 3 ul of the solution was tested by ddPCR. Values obtained were normalized using the two reference genes PP2A and actin. **(B)** Ratio of RNA2 to RNA1 was calculated using the values in (A).

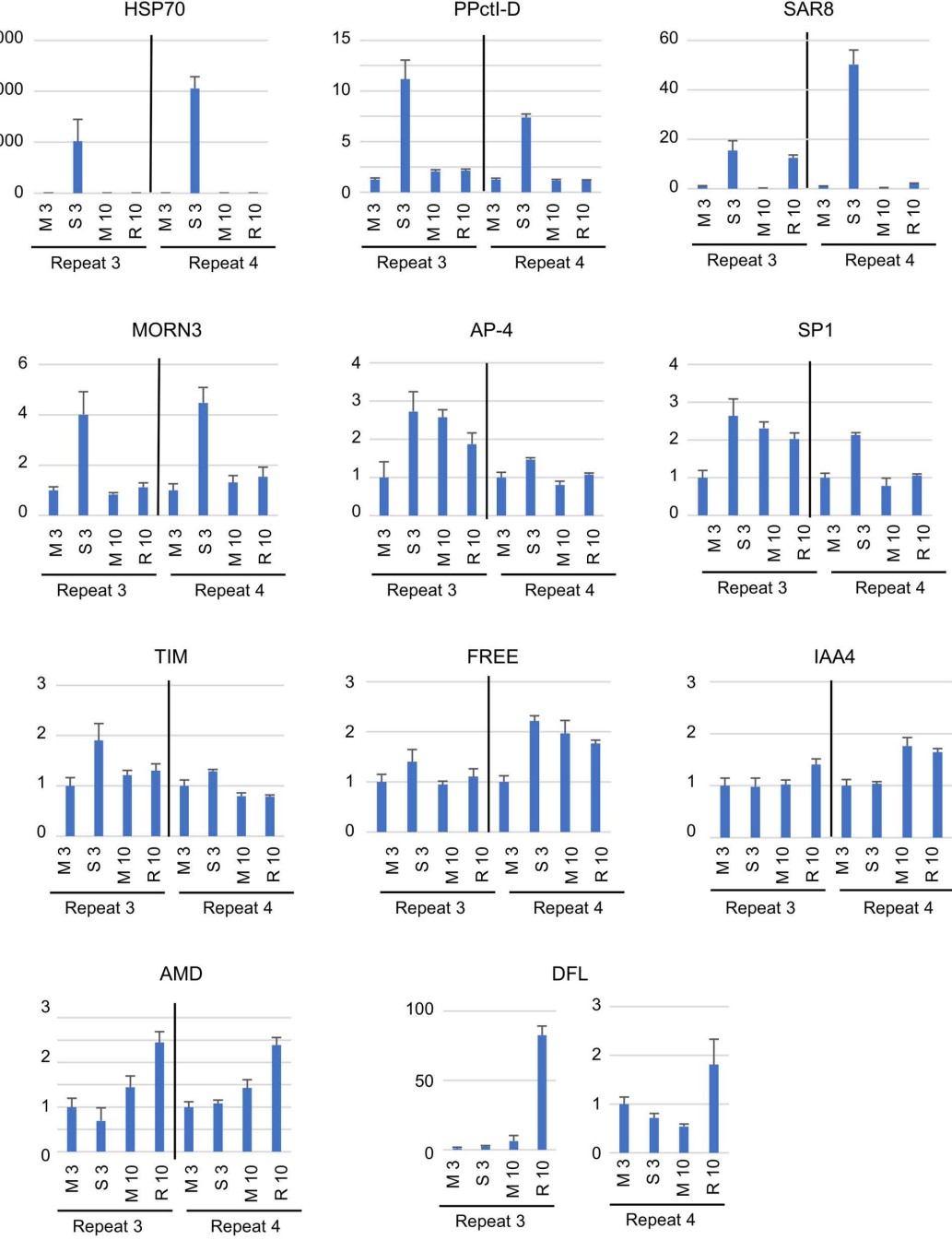

**Fig 7. Quantification of selected plant mRNAs using ddPCR.** Total RNA samples extracted from mock-inoculated leaves at 3 dpi (M 3) or 10 dpi (M 10) or from symptomatic (S 3) or recovered leaves (R 10) were quantified by ddPCR. Total RNAs were diluted 1:20 prior to the ddPCR reaction. Exceptions were DFL (1:5 dilution), HSP70 and PPctI (1:100 dilution for RNAs for S3 samples), and SAR8 (1:200 dilution for RNAs for S3 and R10 samples). Values were normalized using the two reference genes PP2A and actin. Values are reported as fold change compared to the M3 sample (set at 1) for each biological repeat. HSP70: heat shock protein 70; PPctI-D: Peptidyl-prolyl cis-trans isomerase D (cyclophilin); SAR8: mRNA inducible by salicylic acid or by TMV during systemic acquired resistance; MORN3: MORN-motif repeat protein similar to AT1G21920.1 MORN3 (MRF1) protein; AP-4: Transcription factor AP-4; SP1: signal peptidase 1, TIM: Mitochondrial import inner membrane translocase subunit Tim17/Tim22/Tim23 family; FREE: Zinc finger FYVE domain-containing protein 21; IAA4: Auxin-responsive protein IAA4; AMD: Amidotransferase; DFL: defensin-like protein.

gene, IAA4 (Niben101Scf00286g01008, auxin-responsive protein IAA4, a transcription factor), did not show upregulation in either symptomatic or recovered leaves when tested by ddPCR and could therefore not be validated.

The degree and timing of expression for many DEGs varied amongst biological repeats for both the transcriptomic dataset and the ddPCR results. As noted above, we have observed similar differences in the exact timing of early AGO2 induction during repeated ToRSV inoculation experiments in a previous study [30]. We also acknowledge that in the absence of visual symptoms, it is difficult to determine if the leaves collected from the recovered stage of different biological repeats are equivalent in terms of the timing of disease progression, probably explaining some of the discrepancies observed between the ddPCR and transcriptomic results.

## Differential expression of plant miRNAs at different stages of infection

Nine differentially expressed miRNAs were identified in the different comparisons (absolute fold change ≥2 or ≤0.5, p value ≤0.05 and max group mean ≥5.0, Fig 4B; S6 File). Differentially expressed miRNAs were not identified when the two mock samples were compared. Most of the identified miRNAs were only upregulated in the symptomatic vs. mock 3 dpi comparison. However, the expression of three members of the miR391 family, representing the precursor miR391 as well as the two mature cleaved products miR391-5p and miR391-3p continued to increase in recovered leaves. To identify potential targets of the differentially regulated miRNAs, we used the psRNAtarget server and the Niben101 database [92]. Several predicted targets were found in our DEG lists (S6 File).

miR391 was previously shown to be induced in a tolerant interaction between Russet Burbank potato cultivar and a strain of potato virus Y (PVY[N-Wi]) (resulting in an asymptomatic systemic infection) [85]. miR391 targets autoinhibited $Ca^{2+}$-ATPase 10 (ACA10) in *A. thaliana*, and the homologous miR1432 targets a calmodulin-like sensor in rice, both of which are involved in calcium-mediated signalling and likely play roles in plant growth and development as well as immunity [93]. Although neither of these previously identified targets appeared in our DEG lists, we note an EF-hand domain-containing protein in the list of predicted miR391-3p targets, which is upregulated in symptomatic leaves, but not in recovered leaves (S6 File). EF-hand domain-containing proteins bind calcium and are also likely involved in calcium signaling [94]. We also identified several additional potential targets which were down-regulated in symptomatic leaves, including two genes associated with chloroplast function (translation elongation factor Ts, a target of miR391-3p and photosystem I reaction center subunit IV, a target of miR391-5p).

miR530, miR1919, miR397-5p and miR1446 were induced in symptomatic leaves and returned to their basal levels in recovered leaves. miR530-3p and miR530-5p are upregulated during infection of rice by a double-stranded RNA virus, and miR530-5p expression was found to be positively correlated with the expression of its target, plastid-specific 30S ribosomal protein 1 [95]. miRNA530 is also induced following infection with rice blast fungus, and mitigates the resistance of rice to the fungus [96]. In another plant-fungus interaction between cotton and *Verticillium dahlia*, miR530 was shown to be induced by the systemic acquired resistance signal and to cleave an A20/AN1 zinc-finger domain-containing stress-associated protein, resulting in decreased resistance against the fungus [97]. Interestingly, our psRNAtarget analysis also identified "zinc finger A20/AN1 domain-containing stress-assoc. protein 1" as a potential target of miR530 in the ToRSV-*N.benthamiana* interaction (S6 File). This gene is induced during the symptomatic stage of infection but returns to its basal level in recovered leaves, possibly contributing to the attenuation of the HR-like response and of the necrotic symptoms at the recovery stage. A RING finger protein is also predicted as a target of miR530 and has a pattern of expression similar to that of the stress-associated protein. The last two predicted targets are induced in recovered leaves and encode a putative thionin and a receptor-like protein kinase, both of which are likely components of plant defense responses. Although these new targets will need to be validated, these results suggest a complex regulatory network of miR530 with various plant defense responses.

miR1919 showed the highest expression levels amongst miRNAs induced by ToRSV infection. Interestingly, miR1919 was previously implicated in the symptom recovery-related dark green islands phenotype observed in tobacco infected

by tobacco mosaic virus [98]. Similar to our observations, miR1919 was shown to be highly upregulated in mosaic tissues but not in the dark green islands. It is also induced in association with cucumber mosaic virus-induced mosaic symptoms. Overexpression of miR1919 increases susceptibility to both viruses [98]. Amongst predicted miR1919 targets, we found two chaperones that were upregulated in symptomatic leaves (S6 File). The last predicted target was downregulated in symptomatic leaves and was annotated as pseudo-response regulator 7, a transcriptional repressor that regulates the circadian clock and responses to abiotic stresses, including oxidative stress [99].

The expression levels and degree of induction of miR397-5p and miR1446 in symptomatic leaves are modest, compared to miR530 and miR1919, but they could still contribute to the regulation of symptom expression. miR397 is induced by various abiotic and biotic stresses, including virus infection [100–102]. Interestingly, its expression is suppressed by salicylic acid [103]. Since SA marker genes are induced during the symptomatic stage of ToRSV infection [20], it is possible that the progressive accumulation of SA contributes to bringing the levels of miR397 back down after recovery. Overexpression of miR397 results in increased susceptibility to a fungal pathogen [103,104]. Three predicted targets of miR397 were differentially regulated and included two chaperones, peptidyl-prolyl cis-trans isomerase-like 1 and chaperone protein ClpB, which were repressed and induced in symptomatic leaves, respectively (S6 File). In contrast to our results, miR1446 was previously reported to be repressed during abiotic and biotic stresses, including infection with an RNA virus [105,106]. Our analysis did not reveal any potential target for this miRNA in our DEG lists.

miR398 and miR396a only showed differential regulation when comparing recovered leaves to symptomatic leaves (induced and repressed, respectively). Given that levels of expressions were not statistically different in the comparisons of ToRSV-infected leaves to corresponding mock controls, the relevance of these miRNAs is unknown, although a few predicted targets showed differential regulation under various conditions (S6 File).

We have highlighted above various patterns of expression of miRNAs and of their putative targets, including negative or positive correlations as well as temporal regulation of expression following infection with ToRSV. The regulation of target genes by miRNAs is complex and can involve one or several feedback loops, which affect the patterns of expression of the target genes [107]. This is exemplified by the complex regulation of AGO1 homeolog homeostasis by members of the miR168 family [108–110]. In addition, accumulation of VSRs during viral infection can disrupt miRNA repression of many defense-related genes, including AGO1 [111]. Thus, both positive and negative correlations between the expression of miRNAs and their targets can be functionally relevant [107,112]. Although some of our predicted targets were previously validated in other plant-pathogen interactions, most have not been previously described and would require experimental validation. A finer time-course experiment could also help better understand the interaction between the differentially regulated miRNAs and their targets.

## Conclusions

We have examined the transcriptome of ToRSV-infected plants during the symptomatic stage and after symptom recovery. A previous study on transcriptomic changes in *N. benthamiana* plants infected with ToRSV (symptomatic stage) used the microarray technique and reported 1082 DEGs [32]. Although we report slightly fewer DEGs (820) in the symptomatic stage in our study, the enriched functional categories (GO terms) were very similar to the previous study as already discussed above. This is interesting, especially considering that plants were grown under different environmental conditions (27°C in our study vs 22°C in the previous study) and distinct ToRSV isolates were used.

We show that the extent of transcriptomic changes is correlated with symptom intensity, with the majority of genes differentially regulated in symptomatic leaves returning to basal levels in recovered leaves. This is consistent with previous studies that also showed correlations between symptom severity and the number of DEGs [17,32,35]. Genes that were upregulated in recovered leaves (recovered vs mock comparison) belonged to two categories: those that were already upregulated earlier in infection and those that are specific to the recovery stage of infection. Both sets of genes could be relevant to symptom recovery. The list of genes that are upregulated in recovered leaves is diverse and in fact, no specific GO terms were enriched in this comparison. Nevertheless, we noted some interesting trends, in particular the up-regulation of several types

of pathogenesis-related small cysteine-rich proteins (defensins, thionin and nsLTP) in recovered leaves. While some were already upregulated in symptomatic leaves, others are specific to the recovery stage, in particular two defensins. It will be interesting to examine the potential role of these proteins in establishing or maintaining symptom recovery.

We describe the induction of several miRNAs, notably miR391 and miR1919, which were previously described in association with asymptomatic virus infections or with the formation of dark green islands, a phenomenon related to symptom recovery. We also report differential regulation of miR530 and of its previously validated target zinc finger A20/AN1 domain-containing stress-assoc. protein 1, a protein previously shown to be essential for resistance to a fungus in cotton. Several new targets were also predicted for these and other miRNAs which showed various patterns of differential regulation. Many of these predicted targets, particularly for miR530, were related to plant defense responses.

Together these results provide new insights into the transcriptome and miRNA expression changes associated with symptom induction and symptom recovery in *N. benthamiana* plants infected with ToRSV. Further characterization of the DEGs and miRNAs identified in this study may lead to the development of new strategies for mitigating the diseases induced by ToRSV and other nepoviruses.

## Materials and methods

### Plant growth and virus inoculations

The severe ToRSV-Rasp1 isolate was used for all inoculation experiments. *N. benthamiana* plants were grown in Conviron® growth chambers set at 27°C with a 16/8-hour day/night cycle. Two leaves were rub-inoculated in the presence of Carborundum using either virus-infected leaf tissue macerated in phosphate buffer or phosphate buffer only (mock-inoculated control, referred to as "mock control").

### Sample collection and RNA extraction

Symptomatic systemically infected leaves of virus-inoculated plants or equivalent leaves from mock-inoculated plants were harvested at 3 dpi. Upper recovered leaves of virus-inoculated plants and equivalent leaves from mock-inoculated plants were collected at 10 dpi. Leaves from a minimum of three plants were harvested (two leaves/plants) and pooled together to take into account plant to plant variation. Samples were stored at -80°C before total RNA extraction. Total RNA, including small RNA (> 18 nts), were extracted using Qiagen miRNeasy Mini Kit following the manufacturer's recommended protocol. Leaves were ground in liquid nitrogen in a mortar and pestle prior to adding the QIAzol Lysis Reagent. On-column DNase treatment was performed as recommended by the supplier to remove genomic DNA contamination.

### RNA sequencing

Total and small RNA sequencing was carried out using the commercial service provider Applied Biological Materials Inc., Canada (abm). Total RNA samples were checked for quality using an Agilent 2100 Bioanalyzer and an RNA 6000 Pico kit. For total RNA sequencing, plant rRNA was depleted using RiboZero Plant Leaf (Illumina) followed by fragmentations, cDNA synthesis, adenylation of 3' end, adapter ligation and DNA fragment enrichment using the TrueSeq Stranded mRNA LT (Illumina). Samples were run for cluster generation and total RNA sequencing (2x75 bp paired-end) using NextSeq 500 (Illumina). For small RNA sequencing, libraries were prepared using the NEBNext Small RNA Library Prep Kit and sequenced. The first two repeats were conducted in 2016 and sequenced using 2x75 bp. The third repeat was sequenced in 2018 using 75 bp single-end sequencing.

### Analysis of the NGS data

RNA sequences were analyzed using the commercial software CLC Genomics Workbench (version 21). For total RNA, sequenced reads were trimmed for quality using the following parameters: Trim using the quality score (yes), Quality limit

(0.05), Trim ambiguous nucleotide, Maximum number of ambiguities = 2, Automatic read-through adapter trimming (Yes), Remove 5' terminal nucleotides (No), Remove 3' terminal nucleotides (No), Trim to a fixed length (No), Maximum length (150), Trim end (Trim from 3'-end), Discard short reads (No), Discard long reads (No), Save discarded sequences (No), Save broken pairs (No), Create report (Yes).

The draft sequences of the *N. benthamiana* reference genome (v1.0.1) were downloaded from the SolGenomics network website (https://solgenomics.net/) (on 25th May 2021). Trimmed reads were mapped to reference genome using Gene track (Niben101_annotation.allfeatures.wdesc_Gene). Mapped reads were then used for differential gene expression studies. DEGs were selected that displayed an absolute fold change ≥2 or ≤0.5, p value ≤0.05 and max group mean ≥5.0.

AgriGo [113] was used to run SEA. The DEGs were divided into two lists: upregulated and downregulated genes, which were loaded separately into AGRIGO for SEA analysis using the reference genome from the SolGenomics. The portion of genes with GO annotations is indicated in S5 File. These were compared to a list of 31,449 annotated genes from the reference library. For the analysis, the following parameters were selected: Statistical methods – Fisher, Multi-test adjustment methods – Hochberg (FDR), Significance level – 0.05, Minimum number of mapping entries – 5, Gene ontology type: Complete GO. To visualize the SEA analysis, Fig 5 was created in R (v4.4.2) using ggplot2 (v3.52), ggh4x (v0.3.1), and svglite (v2.2.1).

### miRNA analysis

Trimmed small RNAs were run against the miRbase v.22 database (considering known miRNAs from *Nicotiana tabacum*, *Arabidopsis thaliana*, *Solanum lycopersicum*, *Solanum tuberosum*, *Zea mays*, *Triticum aestivum*, *Vitis vinifera* and *Oryza sativa*) to identify conserved miRNAs allowing a maximum mismatch of 2 bases. The trimmed reads (grouped on mature) were annotated with RNAcentral accession numbers. This provides an accession number ID to link the miRbase ID to the RNA central identifier for further downstream analysis. Mature miRNAs were analyzed, and differentially expressed known miRNAs were identified (absolute fold change ≥2 or ≤0.5, p value ≤0.05 and max group mean ≥5.0). Targets of the differentially regulated miRNAs were predicted using the psRNAtarget server (default parameters) and the Niben101 database [92]. Lists of predicted targets were then compared to the lists of DEGs for each comparison to identify targets that were differentially regulated in symptomatic or recovered leaves.

### Mapping of vsiRNAs to the viral genomic RNAs

Trimmed small RNA reads from symptomatic (3 dpi) or recovered (10 dpi) leaf samples were mapped to the ToRSV Rasp1 genomic RNAs (NCBI accessions KM083894 for RNA1 and KM083895 for RNA2) using the following parameters: Reference type = One reference sequence per transcript, use spike-in controls = no, mismatch cost = 2, insertion cost = 2, deletion cost = 3, length fraction = 1, similarity fraction = 1, global alignment = no, strand-specific = both, library type = bulk, the maximum number of hits for a read = 10, count paired reads as two = no, expression value = total counts, minimum read count fusion gene table = 5, create read track, report, fusion gene table, list of mapped reads = yes. Reads per kilobase per million (rpkm) values were then presented in a table. Similar parameters were also used to map trimmed small RNA to 1000-nt fragments of viral sequences.

### Validation of differentially expressed genes by droplet digital PCR

The nucleotide sequences of selected DEGs were first used to search for similar sequences in other *Nicotiana* species within the NCBI database to validate their annotations. These sequences were then aligned using Clustal Omega and primers were designed from conserved regions using the SnapGene software. Absence of predicted off-targets in the *N. benthamiana* reference genome was confirmed for each primer using the CLC search function. Two pairs of ToRSV-specific primers targeting RNA1 and RNA2 were also designed. Two reference genes

(PP2a and Actin) were selected based on a previous study [91] and on their relatively consistent expression in our RNA sequencing data. All primers were synthesized by a commercial provider (IDT, Coralville, Iowa 52241, USA) and are listed in S1 Table.

Total RNA (2 µg) was converted to cDNA using SuperScript™ VILO™ cDNA Synthesis Kit (ThermoFisher Scientific). To prevent saturation of targets in the droplets, cDNA dilutions were adjusted based on preliminary ddPCR data. Quantification of cDNA from the samples was carried out using the QX200 droplet digital PCR system (BioRad). A total of 20 µl reaction volume containing 4 µl of ddPCR EvaGreen Supermix, 1.5 µl of each primer (2.2 µM concentration), 3 µl of diluted cDNA, and water were mixed and used for droplet generation, followed by PCR with a 2.5°C/sec ramp and varying annealing temperatures as optimized for each specific primer pair (see S1 Table). The droplets were read by QX200 Droplet Reader.

## Supporting information

**S1 Fig. Heat map illustrating the hierarchical clustering of samples based on their expression profiles.**
(PPTX)

**S2 Fig. Distribution of virus-derived small RNA on the viral genome.**
(PPTX)

**S3 Fig. Selection of reference genes for ddPCR validation.**
(PPTX)

**S1 Text. Alignment of the nucleotide sequence of RNA1 comparing the original 2014 sequence to that assembled from RNA seq data for each biological repeat (Rep1, Rep2, Rep3).**
(DOCX)

**S2 Text. Alignment of the nucleotide sequence of RNA2 comparing the original 2014 sequence to that assembled from RNA seq data for each biological repeat (Rep1, Rep2, Rep3).**
(DOCX)

**S1 File. List of DEGs for the comparison of symptomatic leaves to mock-inoculated leaves (3 dpi).**
(XLSX)

**S2 File. List of DEGs for the comparison of recovered leaves to mock-inoculated leaves (10 dpi).**
(XLSX)

**S3 File. List of DEGs for the comparison of recovered leaves to symptomatic leaves.**
(XLSX)

**S4 File. List of DEGs for the comparison of mock-inoculated leaves (10 dpi) to mock-inoculated leaves (3 dpi).**
(XLSX)

**S5 File. Lists of enriched GO terms for the symptomatic leaves to mock-inoculated leaves (3dpi) comparison and for the recovered leaves to symptomatic leaves comparison.**
(XLSX)

**S6 File. List of differentially expressed miRNAs and of their differentially regulated predicted targets.**
(XLSX)

**S1 Table. List of primers.**
(DOCX)

## Acknowledgments

We thank Mr. Mark Lubberts (Summerland Research and Development Centre, Agriculture and Agri-Food Canada) for advice and help with bioinformatic analyses and data visualization. We also thank Dr. Basudev Ghoshal (Summerland Research and Development Centre) for helpful discussions.

## Author contributions

**Conceptualization:** Dinesh Babu Paudel, Hélène Sanfaçon.

**Data curation:** Dinesh Babu Paudel, Ana Priscilla Montenegro Alonso.

**Formal analysis:** Dinesh Babu Paudel, Ana Priscilla Montenegro Alonso, Huogen Xiao, Hélène Sanfaçon.

**Funding acquisition:** Hélène Sanfaçon.

**Investigation:** Joan Chisholm, Huogen Xiao.

**Methodology:** Dinesh Babu Paudel, Ana Priscilla Montenegro Alonso, Joan Chisholm, Huogen Xiao.

**Project administration:** Hélène Sanfaçon.

**Supervision:** Hélène Sanfaçon.

**Validation:** Dinesh Babu Paudel, Ana Priscilla Montenegro Alonso, Joan Chisholm, Huogen Xiao, Hélène Sanfaçon.

**Visualization:** Dinesh Babu Paudel, Ana Priscilla Montenegro Alonso, Huogen Xiao, Hélène Sanfaçon.

**Writing – original draft:** Dinesh Babu Paudel, Ana Priscilla Montenegro Alonso.

**Writing – review & editing:** Dinesh Babu Paudel, Ana Priscilla Montenegro Alonso, Joan Chisholm, Huogen Xiao, Hélène Sanfaçon.

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
