## [Decision Letter · Decision Letter 0]

PONE-D-25-21981Transcriptomic changes associated with infection of Nicotiana benthamiana plants with tomato ringspot virus (genus Nepovirus) during the acute symptomatic stage and after symptom recoveryPLOS ONE

Dear Dr. Sanfaçon,

Thank you for submitting your manuscript to PLOS ONE. After careful consideration, we feel that it has merit but does not fully meet PLOS ONE’s publication criteria as it currently stands. Therefore, we invite you to submit a revised version of the manuscript that addresses the points raised during the review process.

We look forward to receiving your revised manuscript.

Kind regards,

Basavaprabhu L Patil, Ph.D.

Academic Editor

PLOS ONE

Journal Requirements:

Agriculture and Agri-Food Canada core funding

We thank Dr. Basudev Ghoshal (Agriculture and Agri-Food Canada) for helpful discussions. This work was supported by Agriculture and Agri-Food Canada core funding.

Agriculture and Agri-Food Canada core funding

Reviewers' comments:

Reviewer's Responses to Questions

**Comments to the Author**

1. Is the manuscript technically sound, and do the data support the conclusions?

Reviewer #1: Yes

Reviewer #2: Yes

Reviewer #3: Yes

2. Has the statistical analysis been performed appropriately and rigorously? 

Reviewer #1: Yes

Reviewer #2: Yes

Reviewer #3: N/A

3. Have the authors made all data underlying the findings in their manuscript fully available?

Reviewer #1: Yes

Reviewer #2: Yes

Reviewer #3: Yes

4. Is the manuscript presented in an intelligible fashion and written in standard English?

Reviewer #1: Yes

Reviewer #2: Yes

Reviewer #3: Yes

5. Review Comments to the Author

Reviewer #1: This manuscript presents a well-executed transcriptomic analysis of ToRSV-infected N. benthamiana, offering novel insights into symptom recovery. While the findings are compelling, additional functional validation and deeper mechanistic exploration would elevate the impact.

Reviewer #2: The manuscript titled “Transcriptomic changes associated with infection of Nicotiana benthamiana plants with tomato ringspot virus (genus Nepovirus) during the acute symptomatic stage and after symptom recovery” presents an interesting and well-structured study that contributes valuable insights into host-virus interactions and transcriptomic dynamics during different stages of ToRSV infection.

However, I have the following major concerns that should be addressed to strengthen the manuscript:

1. Infectivity Data in Figure 1: Please include data on infectivity in Figure 1—specifically, symptomatic infected leaves of virus-inoculated Nicotiana benthamiana plants during the acute stage. This addition will help establish a clearer link between symptom expression and viral presence.

2. Graphical Representation of Gene Set Enrichment Analysis:

I recommend including a visual summary of the Gene Set Enrichment Analysis (GSEA), highlighting the overrepresented gene sets (both upregulated and downregulated) identified in the transcriptome analyses (Table S8).

3. Comparative Table with Tomato ToRSV Data:

It would be beneficial to include a comparative table summarizing transcriptomic responses to ToRSV infection in tomato (from prior studies) alongside the current results in Nicotiana benthamiana. Emphasizing key genes or pathways shared or uniquely regulated in each host would provide important context and increase the impact of the findings.

Minor Concern:

In the abstract, please include the full form of ToRSV (Tomato ringspot virus) at its first mention for clarity, especially for readers less familiar with the abbreviation.

Reviewer #3: Authors have presented data on transcriptome of symptomatic vs recovery sage of Nicotiana benthamiana plants infected with tomato ringspot virus (ToRSV). The extent of changes in the plant transcriptome was correlated with symptom intensity, however, some genes were observed remain upregulated after the symptomatic stage or at the symptom recovery stage. Authors also identified several miRNA and many miRNA predicted targets were related to plant defense responses and may play a potential role for symptom expression and its recovery.

It seems that this pathosystem is highly sensitive to highly on environmental conditions specially few degree changes in temperature can have great impact on virus infection as indicated by numbers of Deg identified in present vs previous reports by others ( Dardick 2007). Do you really think that changes in gene expression/ DEGs were correlated with mainly with temperature other other factor has to play key role? More supporting data and discussion needed for understanding.

It mentioned that serial passage of the virus may results in genetic drift. How many passage were made and the time period for point mutations were observed as motioned in Table 2? Is it a natural mutation? If yes how it was made sure its not sequencing error? What was the impact of these mutations and its interaction with host and disease development? Is it necessary to keep table 2? If not, place in supplementary data.

As I understand, viral nucleotide sequences of 3’ UTR of RNA1 and RNA2 are identical. However, significant differences were shown while mapping of virus derived small RNA for at 3 UTR between RNA1 and RNA2. What were the reasons for this? Figure 2 showing the mapping of virus derived small RNA to the viral genome is not very clear and can be deleted if not adding any specific or significant information.

According to authors, most of the DEGs were return to their basal levels in recovered plants. It is interesting see the virus titer data in just recovered plants with various time gaps.

6. PLOS authors have the option to publish the peer review history of their article (what does this mean? ). If published, this will include your full peer review and any attached files.

**Do you want your identity to be public for this peer review?** For information about this choice, including consent withdrawal, please see our Privacy Policy .

Reviewer #1: **Yes: ** Dr. Madhvi Naresh

Reviewer #2: **Yes: ** Dr. Manojkumar Arthikala

Reviewer #3: No

---

## [Author Response · Author response to Decision Letter 1]

25 Jun 2025

please see "Response to Reviewers" document. Text of the document appear at the end of the amalgamated PDF, after the "Revised Manuscript with track change".

The most relevant responses to Reviewers are also copied below.

We sincerely thank the reviewers for their valuable comments and suggestions, which have helped us improve the manuscript. Below, we address each comment in detail. Please note that line numbers refer to the “Revised manuscript with track change” file.

Several suggestions from the reviewers, such as additional replications, validation of miRNA targets, and functionality studies of DEGs or miRNAs using VIGS or CRISPR, will be valuable for future research, as they will contribute to a deeper understanding of the complexity of plant-nepovirus interaction. However, conducting such experiments would require at least one to two years of experimental work, and are beyond the scope of the “minor revisions” recommended by the editor.

Review Comments to the Author

Reviewer #1:

This manuscript presents a well-executed transcriptomic analysis of ToRSV-infected N. benthamiana, offering novel insights into symptom recovery. While the findings are compelling, additional functional validation and deeper mechanistic exploration would elevate the impact.

Recommendations

1. Address Replicate Variability: Include a supplementary analysis (e.g., hierarchical clustering or batch effect correction) to confirm that biological conclusions are robust despite technical differences between replicates.

We have conducted a hierarchical clustering analysis (S1 Fig) and added comments in the text (line 177-182). This new analysis confirms the variability between replicates already observed with the PCA analysis. Although the distinct patterns of the third repeat are evident in the generated heat map, a large number of common DEGs shared amongst all symptomatic samples (from all three repeats) are also observed.

We acknowledge that the observed variability amongst our replicates complicates the interpretation of the results. However, it is important to note that the results described in this study are consistent with results observed in previous studies. In particular, the functional categories of DEGs identified in symptomatic leaf samples in this study were very similar to those identified in a previous study (Dardick, 2007; reference 32 in the manuscript), even though the ToRSV strain, the environmental conditions and the technology used to identify the DEGs were different. As highlighted in the manuscript, the results of this study are also consistent with several of our previous studies in which we had already observed down-regulation of photosynthesis-associated genes, and transient up-regulation of AGO2 in symptomatic leaves (Ghoshal et al, 2014; Paudel et al, 2018; references 18 and 30). In addition, we were able to validate several new DEGs in this study using ddPCR, notably the defensins associated with the symptom recovery stage.

Validate Key Findings: Perform functional assays (e.g., VIGS or CRISPR) to test the role of defensins or miR530 targets in symptom recovery. Validate miRNA-target interactions experimentally (e.g., dual-luciferase assays).

These are excellent suggestions for continuing studies to further elucidate the exact mechanism of symptom recovery. However, these suggestions would require extensive experimental work (likely several years), and are beyond the scope of the “minor revisions” recommended by the reviewer. For example, we previously used VIGS and CRISPR-induced mutations to examine the role of AGO2 in ToRSV infection (Paudel et al, 2018; reference 30). Although we had observed a peak of AGO2 expression at early stages of infection with all ToRSV isolates and environmental conditions tested, the accumulation of AGO2 protein was only observed under certain conditions, revealing complex post-transcriptional regulation of this gene. Furthermore, while the accumulation of AGO2 protein at early stages of ToRSV infection was clearly correlated with the establishment of symptom recovery later on, silencing of AGO2 by VIGS or mutation of AGO2 by CRISPR did not prevent symptom recovery, although it resulted in increased virus accumulation. This previous study highlights the complexity of validating the roles of identified genes or miRNA in planta.

We agree with the reviewer that many proposed miRNA-targets will require validation in the future. As discussed in the text, the zinc finger A20/AN1 protein target of miR530 was already validated in a previous study (in the context of a plant-fungus infection) (Hu et al 2023, reference 97). We have clarified throughout the text that other targets are only predicted at this stage.

We acknowledge that this study constitutes a first step in the characterization of plant-nepovirus interactions, but firmly believe that the identification of genes and miRNAs that are differentially regulated at various stages of infection will help guide future research.

2. Expand Discussion: Elaborate on the potential mechanisms by which defensins or miRNAs mediate recovery (e.g., antimicrobial activity, signaling cross-talk). Discuss how chloroplast suppression during symptoms might synergize with defense responses.

We have added text to elaborate on the potential role of defensin (lines 403 to 412). The role of miRNA in plant-virus interactions is complex, as many defense-response genes are under the control of miRNAs, the regulation of which can be affected by viral suppressors of RNA silencing. We have extensively discussed the possible roles of miRNAs in the “Differential expression of plant miRNAs at different stages of infection” section of the Result and Discussion and have added a sentence to explain how VSRs can disrupt miRNA regulation (lines 611-613) to provide clarification. Similarly, we updated the text (lines 363, 365-366) on the role of the chloroplast in symptom development.

Clarify Methods: Specify how "equivalent leaves" were selected during sampling to ensure consistency. Provide more details on small RNA library preparation (e.g., adapter sequences, size selection).

We have clarified the text (lines 128-133, and 137-138 and have included a figure (new Fig 1) to illustrate the sampling method and explain how we selected equivalent leaves.

RNA library preparation and sequencing were outsourced to a commercial provider (Applied Biological Materials, Canada). The company provided information stating that the small RNA library construction was performed using the NEBNext Small RNA Library Prep kit, following the manufacturer's instructions, as outlined in Materials and Methods, and the detailed protocol is available on the NEB website (neb.com).

Reviewer #2:

The manuscript titled “Transcriptomic changes associated with infection of Nicotiana benthamiana plants with tomato ringspot virus (genus Nepovirus) during the acute symptomatic stage and after symptom recovery” presents an interesting and well-structured study that contributes valuable insights into host-virus interactions and transcriptomic dynamics during different stages of ToRSV infection.

However, I have the following major concerns that should be addressed to strengthen the manuscript:

1. Infectivity Data in Figure 1: Please include data on infectivity in Figure 1—specifically, symptomatic infected leaves of virus-inoculated Nicotiana benthamiana plants during the acute stage. This addition will help establish a clearer link between symptom expression and viral presence.

We added a figure (new Fig. 1) depicting typical symptoms on the inoculated leaves, systemically infected symptomatic leaves, and recovered leaves of ToRSV-infected plants. These symptoms were already described extensively in previous publications, which we are citing. The figure also exemplifies how we have selected equivalent leaves from mock-inoculated plants. We modified the text in the results and discussion section (lines 128-133, and 137-138) to provide additional details on the type of symptoms analyzed and the sampling method.

2. Graphical Representation of Gene Set Enrichment Analysis:

I recommend including a visual summary of the Gene Set Enrichment Analysis (GSEA), highlighting the overrepresented gene sets (both upregulated and downregulated) identified in the transcriptome analyses (Table S8).

We have added a new figure to provide a visual summary of the results (new Fig 5, see also figure legend-lines 293-295). For the figure, we focused on the categories “molecular function” and “biological processes” which are discussed in the text. S5_File (previously labelled as supplemental S8) is unchanged and is available to the readers interested in a deeper exploration.

3. Comparative Table with Tomato ToRSV Data:

It would be beneficial to include a comparative table summarizing transcriptomic responses to ToRSV infection in tomato (from prior studies) alongside the current results in Nicotiana benthamiana. Emphasizing key genes or pathways shared or uniquely regulated in each host would provide important context and increase the impact of the findings.

To our knowledge, transcriptomic responses to ToRSV infection have not been studied in tomato. The only available prior study analyzed ToRSV infection in N. benthamiana and focused on symptomatic leaves only (Dardick, 2007, reference 32). The study was conducted in 2007, at a time when the N. benthamiana genome was not well annotated and identification of genes was based on comparisons to the potato genome. We attempted to compare genes identified in the two studies but encountered difficulties in identifying the correct homologues in N. benthamiana, because of the advances in the genome annotations. Although it is difficult to make exact comparisons, we note similarities in the annotations of DEGs in the two studies. In particular, the induction of genes categorized as being involved in biotic stresses, protein folding, and disease resistance-related proteins and the repression of genes that are related to photosynthesis and chloroplast translation. This was already discussed extensively in the text.

Minor Concern:

In the abstract, please include the full form of ToRSV (Tomato ringspot virus) at its first mention for clarity, especially for readers less familiar with the abbreviation.

We updated the text as suggested (line 22).

Reviewer #3:

Authors have presented data on transcriptome of symptomatic vs recovery sage of Nicotiana benthamiana plants infected with tomato ringspot virus (ToRSV). The extent of changes in the plant transcriptome was correlated with symptom intensity, however, some genes were observed remain upregulated after the symptomatic stage or at the symptom recovery stage. Authors also identified several miRNA and many miRNA predicted targets were related to plant defense responses and may play a potential role for symptom expression and its recovery.

It seems that this pathosystem is highly sensitive to highly on environmental conditions specially few degree changes in temperature can have great impact on virus infection as indicated by numbers of Deg identified in present vs previous reports by others ( Dardick 2007). Do you really think that changes in gene expression/ DEGs were correlated with mainly with temperature other other factor has to play key role? More supporting data and discussion are needed for understanding.

We agree with the reviewer that many factors could have impacted the number of DEGs identified. These include environmental conditions (22C in the previous study vs 27C in our study), choice of isolate (a mild peach yellow bud mosaic isolate in the previous study vs a severe raspberry isolate in our study), timing of sample collection for symptomatic leaves (14 dpi in the previous study vs 3 dpi in our study) and the method for DEG identification (microarray in the previous study vs transcriptomic in our study). While temperature could have played a role in the number of DEGs identified, it is probably not the only factor. We have removed the sentence suggesting that temperature was the main factor contributing to the differences in the number of DEGs (lines 625-627).

It mentioned that serial passage of the virus may results in genetic drift. How many passage were made and the time period for point mutations were observed as motioned in Table 2? Is it a natural mutation? If yes how it was made sure its not sequencing error? What was the impact of these mutations and its interaction with host and disease development? Is it necessary to keep table 2? If not, place in supplementary data.

Because infectious clones for ToRSV are not available, the virus is routinely maintained in our lab by routine serial passage (rub-inoculation) in Nicotiana benthamiana. The virus was initially sequenced in 2014. Unfortunately, we do not know the exact number of serial passages that occurred between 2014 and the start of our experiment or between each biological repeat. Although we did not observe obvious differences in symptomatology during passaging of the virus, we are aware that serial passaging could cause genetic drift. For this reason, we examined the sequence of the two ToRSV RNAs in each biological repeat. We generated a consensus sequence for the viral RNAs at each biological repeat using combined reads obtained from small RNA and total RNA sequencing (updated text at line 189). Thus, we consider it unlikely that the mutations observed would be the result of sequencing or assembly mistakes. We would also like to point out that the four mutations first reported in the first repeat are consistently seen in the second and third repeats. Likewise, the mutation first noted in the second repeat is also present in the third repeat (we have added this in the text, line 194). If the mutations were the results of random sequencing errors, they would not be expected to reappear in subsequent biological repeats.

In the absence of an infectious clone to conduct targeted reverse-genetic experiments, the potential biological impact of the mutations on the viral infection cycle remains unknown. The four mutations that occurred before the onset of these experiments (mutations first observed in repeat 1) would affect all repeats equally. Of the remaining four mutations, two are synonymous and two are located in the 3’ UTR region, thus none of these mutations would impact the sequence of the viral proteins. Previously, we have shown the involvement of certain regions of ToRSV 5’ and 3’ UTRs in translation enhancement, but the mutations are not located in these regions.

We believe that the information presented in Table 2 is important to provide readers with an understanding of the limited genetic drift that occurred during the course of the experiments (three biological repeats conducted over 2 years).

As I understand, viral nucleotide sequences of 3’ UTR of RNA1 and RNA2 are identical. However, significant differences were shown while mapping of virus derived small RNA for at 3 UTR between RNA1 and RNA2. What were the reasons for this? Figure 2 showing the mapping of virus derived small RNA to the viral genome is not very clear and can be deleted if not adding any specific or significant information.

We have updated the text to better explain that although the 3’ UTRs share extensive stretch of sequence identity, they are not identical (lines 225-230, also line 241 in Fig legend). In the case of the ToRSV-Rasp1 isolate, the 3’ UTRs share 81% sequence identity, unequally distributed between long stretches of nearly identical sequences interspersed by more divergent sequences. This probably explains why the mapping of vsiRNAs to the two 3’ UTRs are both similar and distinct.

The figure (previously Fig 2, now relabeled as Fig 3) provides a snapshot of small RNA distribution on the viral RNA and their changes during the symptomatic and recovery stages. Identification of vsiRNAs hotspot in the genome suggests preferential target sites of the host silencing machinery which could be useful in the future for the design of antiviral strategies. For this reason, we prefer to keep the figure in the core of the manuscript. We have added a sentence in the text (lines 224-22

---

## [Editor Report · Decision Letter 1]

Transcriptomic changes associated with infection of Nicotiana benthamiana plants with tomato ringspot virus (genus Nepovirus) during the acute symptomatic stage and after symptom recovery

PONE-D-25-21981R1

Dear Dr. Sanfaçon,

We’re pleased to inform you that your manuscript has been judged scientifically suitable for publication and will be formally accepted for publication once it meets all outstanding technical requirements.

Kind regards,

Basavaprabhu L Patil, Ph.D.

Academic Editor

PLOS ONE
---

## [Editor Report · Acceptance letter]

PONE-D-25-21981R1

PLOS ONE

Dear Dr. Sanfaçon,

I'm pleased to inform you that your manuscript has been deemed suitable for publication in PLOS ONE. Congratulations! Your manuscript is now being handed over to our production team.

Kind regards,

on behalf of

Dr. Basavaprabhu L Patil

Academic Editor

PLOS ONE